# High-throughput mapping of single-neuron projection and molecular features by retrograde barcoded labeling

Peibo Xu[1,2]*[†], Jian Peng[3†], Tingli Yuan[1†], Zhaoqin Chen[1†], Hui He[1,2†], Ziyan Wu[1], Ting Li[3], Xiaodong Li[1,2], Luyue Wang[4], Le Gao[1], Jun Yan[1,5,6], Wu Wei[4,7], Chengyu T Li[1,5,6,7]*, Zhen-Ge Luo[3]*, Yuejun Chen[1,5]*

[1]Institute of Neuroscience, State Key Laboratory of Neuroscience, Chinese Academy of Sciences, CAS Center for Excellence in Brain Science and Intelligence Technology, Shanghai Center for Brain Science and Brain-Inspired Technology, Shanghai, China; [2]University of Chinese Academy of Sciences, Beijing, China; [3]School of Life Science and Technology & State Key Laboratory of Advanced Medical Materials and Devices, ShanghaiTech University, Shanghai, China; [4]CAS Key Laboratory of Computational Biology, Shanghai Institute of Nutrition and Health, University of Chinese Academy of Sciences, Chinese Academy of Science, Shanghai, China; [5]Shanghai Center for Brain Science and Brain-Inspired Intelligence Technology, Shanghai, China; [6]School of Future Technology, University of Chinese Academy of Sciences, Beijing, China; [7]Lingang Laboratory, Shanghai, China

*For correspondence:
michaelxupb@gmail.com (PX);
tonylicy@lglab.ac.cn (CTL);
luozhg@shanghaitech.edu.cn
(Z-GL);
yuejunchen@ion.ac.cn (YC)

†These authors contributed
equally to this work

Competing interest: The authors
declare that no competing
interests exist.

Reviewing Editor: Jeremy J
Day, University of Alabama at
Birmingham, United States

**Abstract** Deciphering patterns of connectivity between neurons in the brain is a critical step toward understanding brain function. Imaging-based neuroanatomical tracing identifies area-to-area or sparse neuron-to-neuron connectivity patterns, but with limited throughput. Barcode-based connectomics maps large numbers of single-neuron projections, but remains a challenge for jointly analyzing single-cell transcriptomics. Here, we established a rAAV2-retro barcode-based multiplexed tracing method that simultaneously characterizes the projectome and transcriptome at the single neuron level. We uncovered dedicated and collateral projection patterns of ventromedial prefrontal cortex (vmPFC) neurons to five downstream targets and found that projection-defined vmPFC neurons are molecularly heterogeneous. We identified transcriptional signatures of projection-specific vmPFC neurons, and verified *Pou3f1* as a marker gene enriched in neurons projecting to the lateral hypothalamus, denoting a distinct subset with collateral projections to both dorsomedial striatum and lateral hypothalamus. In summary, we have developed a new multiplexed technique whose paired connectome and gene expression data can help reveal organizational principles that form neural circuits and process information.

## Editor's evaluation

This manuscript describes a valuable new circuit mapping and profiling technique called Multiplexed projEction neuRons retrograde barcodE (MERGEseq) that combines transcriptome and projectome data at a single-cell resolution. The authors provide solid evidence that MERGEseq can be used to identify projection targets and cell type/layer/transcriptome differences of projection neurons in the mouse prefrontal cortex, and validation experiments are rigorous. While this report is a proof-of-principle that MERGEseq is useful for circuit mapping and profiling and many potential details will influence conclusions, this technique could easily be adapted to other regions with known projection targets and adds to a growing arsenal of combinatorial circuit mapping and profiling tools.

## Introduction

Wiring diagrams of a brain can be divided into three levels: (1) the macroscale connectome that describes inter-areal connections, (2) the mesoscale connectome that describes connections between cells, and (3) the microscale connectome that describes connections at the synaptic level (*Zeng, 2018*). Studying circuit architecture at the level of the mesoscale connectome describes how information flows between brain regions (*Oh et al., 2014*). Traditionally, neuroanatomical tracers are used to characterize regional connectivity matrices (*Cowan, 1998*). To obtain cell-type-specific connectivity, one can use recombinant virus-based tracer in transgenic model organisms or more precisely trace a specific component of a neural circuit using viral-genetic tracing tools to dissect the input-output organization (*Ghosh et al., 2011*; *Nassi et al., 2015*; *Schwarz et al., 2015*). However, these methods are highly reliant on complex recombinant virus design and genetically modified model organism, and often are not at a single-neuron resolution.

While recent advances have brought invaluable insights into understanding neuronal circuits at single-neuron resolution, existing methods have limitations. High-throughput fluorescence imaging, such as fluorescence micro-optical sectioning tomography (fMOST), can reconstruct detailed neuron morphologies but requires specialized expertise and equipment and lack transcriptomic information (*Gong et al., 2016*; *Rompani et al., 2017*). Barcode-based methods like MAPseq, BRICseq (multiplexed MAPseq), BARseq, and ConnectID utilize sequencing to map projections (*Chen et al., 2019*; *Huang et al., 2020*; *Kebschull et al., 2016*; *Klingler et al., 2021*). However, MAPseq and BRICseq can only provide connectome information (*Huang et al., 2020*; *Kebschull et al., 2016*), BARseq is constrained to assessing a handful of genes via in situ hybridization (*Chen et al., 2019*), and ConnectID has low recovery of cells with dual connectome-transcriptome data (~16%, 391 cells with connectome barcode identity in 2450 cells with scRNA-seq; *Klingler et al., 2021*). VECTORseq, a Retro-seq-based method (*Tasic et al., 2018*), is limited by its number of transgenic barcodes used (*Cheung et al., 2021*). The updated BARseq protocol enables detection of up to 100 genes, but throughput remains lower and oligo synthesis costs remain higher compared to scRNA-seq (*Chen et al., 2023*; *Sun et al., 2021*). In summary, despite significant progress, existing methods fall short in efficiently integrating high-throughput projectomes and transcriptomes at the single-neuron level, hindering a comprehensive understanding of the connectomic and transcriptomic interplay in neuronal circuitry.

Medial prefrontal cortex (mPFC) is an intricate brain region involved in higher order cognitive functions, information processing (e.g., memory and emotions) and driving goal-directed actions (*Le Merre et al., 2021*). For example, mPFC neurons projecting to the nucleus accumbens encoding punishment-related internal states were located in more superficial layer 5a, and mPFC neurons projecting to the ventral tegmental area encoding aversive learning were located in deeper layer 5b (*Kim et al., 2017*; *Wu et al., 2021*). Although previous studies have extensively investigated the anatomical and functional diversities of mPFC, the relationship between anatomical and molecular features of mPFC neurons remains elusive. Do mPFC neurons projecting to different downstream brain regions differ in their transcriptomes? Are these projection-defined mPFC neurons homogeneous or composed of different neuron subtypes? The answer to these questions may be further complicated by the finding that mPFC neurons can send collateral axons to multiple brain regions (*Cornwall and Phillipson, 1988*). So, what are the principles of target selection or target combination for these collateral projection mPFC neurons? What are the cell type and molecular features of these 'broadcasting' neurons?

To address these challenges, we designed a multiplexed tracing method capable of characterizing single-neuron transcriptome and projectome at the same time, which we called MERGE-seq (Multiplexed projEction neuRons retroGrade barcodE). We used MERGE-seq to interrogate the projectome and the corresponding transcriptome of ventral mPFC neurons. We injected five rAAV2-retro viruses with distinct barcodes into the five known downstream targets of ventromedial prefrontal cortex (vmPFC), including agranular insular cortex (AI), dorsomedial striatum (DMS), basolateral amygdala (BLA), mediodorsal thalamic nucleus (MD), and lateral hypothalamus (LH), in the same mouse brain such that each target region received a unique barcoded rAAV2-retro. We found that vmPFC neurons projecting to each downstream target are heterogeneous, which are composed of transcriptionally different subtypes of neurons. Approximately 65% of barcoded vmPFC neurons exhibited dedicated projection patterns based on MERGE-seq data, sending axonal projections exclusively to one of the five selected targets. It is important to note that this characterization of 'dedicated projection'

neurons is specifically defined in the context of the five target regions examined in this study. Approximately 35% of barcoded vmPFC neurons sent collateral projections to multiple brain regions, most of which are dual-target projection neurons (bifurcated projection). We further uncovered the cell type compositions and layer distributions of these dedicated and collateral projection vmPFC neurons, and revealed their molecular signatures. We validated complex MERGE-seq-inferred projection patterns by joint analysis with recently published single-neuron projectome data (*Gao et al., 2022*). Additionally, dual-modal interrogation using RNA fluorescence in situ hybridization (FISH) and dual-color retrograde AAV labeling allowed us to confirm vmPFC neuron bifurcations to DMS and LH, demonstrating layer 5 *Pou3f1*[+] neurons collateralize between these targets. Finally, we implemented a machine learning-based methodology and uncovered specific gene clusters for predicting certain projection patterns. As MERGE-seq bridges the gap between single-neuron projectome and transcriptome data, it can uncover new molecular properties of anatomical neural circuits.

## Results

### MERGE-seq characterizes single neuron transcriptome and projectome simultaneously

In order to use the 10x Genomics scRNA-seq system to analyze transcripts from cells infected with rAAV2-retro virus, we modified the viral vector by adding a 15 bp barcode index and polyadenylation signal sequences to the 3' end of the EGFP sequences, which was driven by a short CAG promoter (*Figure 1A and B*, see Materials and ethods). Then, five rAAV2-retro viruses with different barcodes were individually injected into five brain regions of the same mouse, including AI, DMS, BLA, LH, and MD. These brain areas are the known downstream brain regions of vmPFC (*Hunnicutt et al., 2016*; *Hurley et al., 1991*; *Reppucci and Petrovich, 2016*; *Vertes, 2004*; *Zhu et al., 2020*). A period of six weeks was set to allow efficient retrograde labeling of vmPFC neurons by these barcoded viruses. These mice were then sacrificed and the vmPFC (specifically the prelimbic area [PrL] and the infralimbic area [IL]) was carefully dissected for scRNA-seq analysis (*Figure 1A*). Single-cell transcriptional libraries were obtained using 10x Genomics library preparation protocols, and virus barcode expression libraries were obtained using user-defined primers, which could enrich cDNA fragments composed of barcode index, unique molecular identifiers (UMIs), and the cell barcode (*Figure 1B*). We detected 24,788 cells in the raw data matrix. Following initial quality control, which ensured the number of detected RNA in each cell ranged between 500 and 8000, RNA UMI counts in each cell were within 1000–60,000, and the percentage of mitochondrial genes remained below 20%, we recovered 1791 cells undergoing fluorescence-activated cell sorting (FACS) from three mice and 19,470 single cells without sorting from the other three mice, a total of 21,261 cells. Transcriptional profiling of all cells revealed major cell types including excitatory neurons (*Slc17a7*[+]), microglia (*C1qa*[+]), endothelial cells (*Itm2a*[+], Endo), oligodendrocyte progenitor cells (*Olig2*[+]*Mog*[-], OPCs), oligodendrocyte (*Olig2*[+]*Mog*[+], Oligo), inhibitory neurons (*Gad1*[+]), astrocyte (*Aldh1l1*[+], Astro), and activated microglia (*C1qa*[+]*Pf4*[+], Act. Microglia) as previously reported (*Bhattacherjee et al., 2019*; *Figure 1C–E*). Barcoded cells below refer to a collection of barcoded cells from unsorted group and FAC-sorted group.

First, we validated that each target region was labeled and effectively covered by the rAAV2-retro-EGFP (*Figure 1—figure supplement 1A*). Next, we showed that there were sufficient sequence differences to distinguish one barcode from others and sufficient sequences difference to identify the right barcode among five references during sequencing (*Figure 1—figure supplement 1B, C*). Since each downstream brain region of vmPFC received a unique and predetermined barcoded virus, each virus barcode identified in a vmPFC neuron represents the specific corresponding downstream brain region that the neuron projects to. We found abundant zero counts for projection barcodes in scRNA-seq libraries, contrasting robust detection in projectome libraries generated by targeted amplification from full-length cDNA (*Figure 1—figure supplement 1D*). To determine validly barcoded cells, we first calculated the 95th nUMI percentile across all barcodes and removed outlier cells with exceptionally high nUMI (see Materials and methods). We used 'EGFP-negative' FAC-sorted cells (defined by nUMI EGFP = 0) and non-neuronal cells from scRNA-seq as negative controls to calculate 99.9th percentile UMI thresholds per barcode using empirical cumulative distribution functions (ECDF; *Figure 1—figure supplement 1E*). By taking the higher threshold for each barcode from these two negative control analyses, we classified cells exceeding these values as validly barcoded. It is worth mentioning

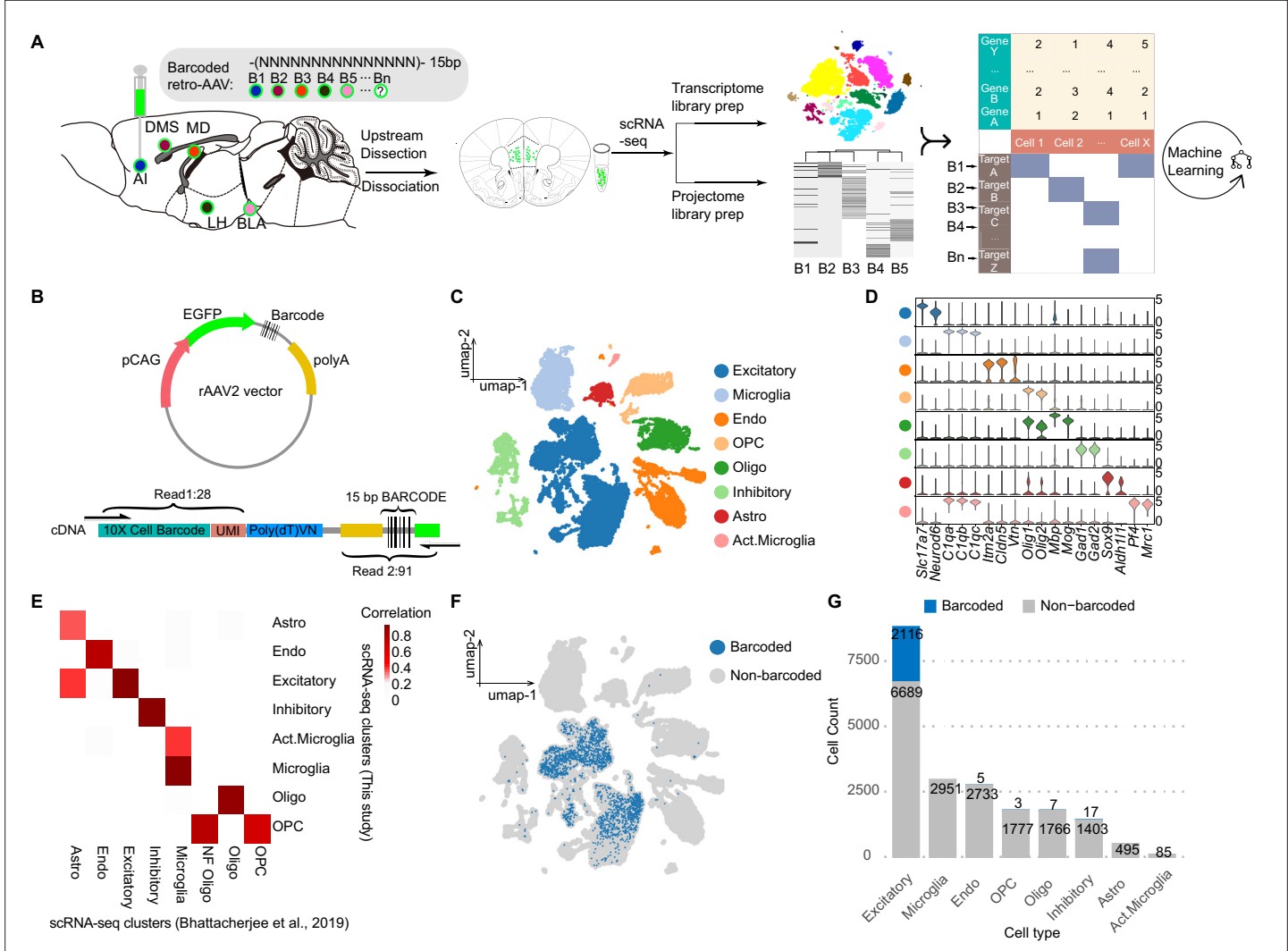

**Figure 1.** MERGE-seq characterized single-neuron transcriptomes and projectomes simultaneously. (**A**) Schematic diagram of the experimental workflow. (**B**) rAAV2 plasmid vector design, and schematic of designed primers to recover cell barcode and UMI in read 1, and 3' tail of EGFP and virus barcode in read 2. According to the recommendation of 10x Genomics, a faithful mapping should cover 28 bp for read 1 and 91 bp for read 2. In our design, 150 bp pair-end sequencing can sufficiently meet the need to recover cell barcode, UMI and virus barcode. (**C**) Umap embedding of transcriptional clustering results for all vmPFC cells. (**D**) Stacked violin plots showing the expression of markers for each cluster. (**E**) Heatmap showing the gene-expression correlation between major cell types defined by scRNA-seq of this study and **Bhattacherjee et al., 2019**. (**F**) Umap embedding of all determined barcoded cells labeled in blue. (**G**) Bar plot showing frequency of barcoded (blue) and non-barcoded (grey) cells in all recovered cell types. In (**C–E**), 21,261cells were represented. In (**F, G**), 20,047cells were represented. 1214 cells with exceptionally high nUMIs were removed.

The online version of this article includes the following figure supplement(s) for figure 1:

**Figure supplement 1.** Validation of rAAV2-retro injection sites and determination of valid barcoded cells.

that the UMI threshold differs for different targets due to different magnitude of barcode expression of each projection target (**Figure 1—figure supplement 1F**). Across all detected cell types, barcoded cells were primarily excitatory neurons rather than inhibitory neurons or non-neuronal cell types (2116 validly barcoded in 8805 excitatory neurons, and 5 validly barcoded in 2738 endothelial cells, 3 validly barcoded in 1780 oligodendrocyte progenitor cells, 7 validly barcoded in 1773 oligodendrocytes, and 17 in 1420 inhibitory neurons, **Figure 1F and G**). This is consistent with the finding that mPFC projection neurons are excitatory (**Gabbott et al., 2005**). Using this stringent threshold, 49.0% of FAC-sorted and 18.7% of unsorted cells were classified as barcoded (**Figure 1—figure supplement 1G**). In parallel, we calculated EGFP$^+$ ratios (nUMI of *EGFP* RNA >0) as 81% for FAC-sorted and 26% on average for unsorted cells (**Figure 1—figure supplement 1H**). The lower fraction of barcoded versus

EGFP⁺ cells suggests our conservative threshold increases false negatives, classifying some low UMI cells as non-barcoded. Therefore, we focused analyses on reliably barcoded cells, though conclusions may not capture the full heterogeneous projection repertoire. Together, these results demonstrate that MERGE-seq can record single neuron transcriptome and projectome simultaneously.

## MERGE-seq reveals transcriptomic heterogeneity and cell type composition of vmPFC neurons projecting to different targets

Previous studies have shown that vmPFC neurons project to multiple brain regions including AI, DMS, BLA, LH, and MD; however, the cell type composition of these projection neurons remains largely unknown (*Le Merre et al., 2021*). Combining with single neuron transcriptome, we explored the transcriptome and subtype composition of vmPFC neurons projecting to different downstream brain regions. We first re-clustered excitatory projection neurons expressing *Slc17a7* (also known as vesicular glutamate transporter, *Vglut1*). Clusters with low gene/UMI counts and high mitochondrial gene expression were filtered out as low-quality (*Ilicic et al., 2016*). Some clusters exhibited non-neuronal cell markers like microglial genes (*C1qa*, *C1qb*), oligodendrocyte genes (*Olig1*, *Olig2*), and endothelial cell genes (*Flt1*, *Cldn5*) despite small cluster size, indicating contamination from other cell types incorrectly grouped within excitatory neurons after initial clustering. In total, we filtered out 637 cells that were identified as either low-quality or contaminated with non-neuronal cell types and recovered 9368 excitatory neurons (see Materials and methods, *Figure 2—figure supplement 1A*, *Supplementary file 1*). We generated seven excitatory neuron clusters, which were annotated based on typical markers of cortical layers (*Bhattacherjee et al., 2019*; *Sorensen et al., 2015*; layer 2/3, *Cux2*; layer 5, *Etv1*; layer 6, *Sulf1*) and differentially expressed genes (DEGs; *Supplementary file 2*). These neuron clusters include L2/3-Calb1 (4.1%), L2/3-Rorb (5.9%), L5-Bcl6 (3.3%), L5-Htr2c (3.9%), L5-S100b (11.6%), L6-Npy (12.6%), and L6-Syt6 (58.7%; *Figure 2A*). The layer and subtype marker genes of these clusters were confirmed to be expressed in corresponding layers in the vmPFC, as revealed by in situ hybridization results of the Allen Mouse Brain Atlas (*Figure 2A*, *Figure 2—figure supplement 1A–C*). Of note, we captured more layer 6 neurons than superficial layer neurons (12.6% L6-Npy and 58.7% L6-Syt6, *Figure 2B*), which is different from a previous report (*Bhattacherjee et al., 2019*). We speculate that different dissociation protocols may cause biased neuron capture.

Cells that were retrogradely barcoded spanned all layers of the vmPFC (layer 2/3, 5, and 6) and included all seven neuronal subtypes (*Figure 2A–C*). These subtypes were highly corresponding to the spatially resolved PFC excitatory neuronal subtypes (*Bhattacherjee et al., 2023*; see Materials and methods, *Figure 2D*). High correlation allows us to infer the spatial localization of our annotated subtypes detected in scRNA-seq data. We also found that excitatory neuronal subtypes are transcriptionally similar to those previously reported (*Figure 2—figure supplement 1D*; *Bhattacherjee et al., 2019*; *Lui et al., 2021*; *Yao et al., 2021*). All these integrated analyses suggest that multiple viral infections will not significantly affect the transcriptional state of these retrogradely labeled vmPFC neurons. For the L5-Htr2c subtype, only nine neurons were recovered with valid barcodes, possibly due to cell loss during single-cell dissociation or tropism of AAV2-retro, or these neurons may intrinsically not project to any target we chose (*Figure 2B*). Neurons projecting to DMS were abundant (n=1242), whereas neurons projecting to BLA were rare (n=163; *Figure 2E*). These results are consistent with data acquired via conventional fluorescence-based retrograde tracing in the prefrontal cortex of rats (*Gabbott et al., 2005*).

We next calculated the subtype composition of vmPFC neurons projecting to each downstream brain region. Interestingly, we found that these target specific projection neurons were transcriptionally heterogeneous, which were composed of different neuronal subtypes (*Figure 2F*). Neurons projecting to LH or MD were mainly L6-Syt6 subtype, whereas neurons projecting to AI, DMS, or BLA were mainly composed of L5-S100b, and to a lesser extent L6-Npy and L2/3-Rorb subtypes (*Figure 2F*). It is worth noting, based on the observed ratios, that the cellular composition of target-specific projection neurons from FAC-sorted or unsorted groups is similar (*Figure 2—figure supplement 1E*).

As the layer distribution of each neuron subtype can be inferred by their layer specific marker gene expression, these results also implied the layer distribution of neurons projecting to each target (*Figure 2E*, *Figure 2—figure supplement 1B*). By calculating the projection properties of each vmPFC neuron subtype, we found that each transcriptome-defined neuron subtype can project to specific but

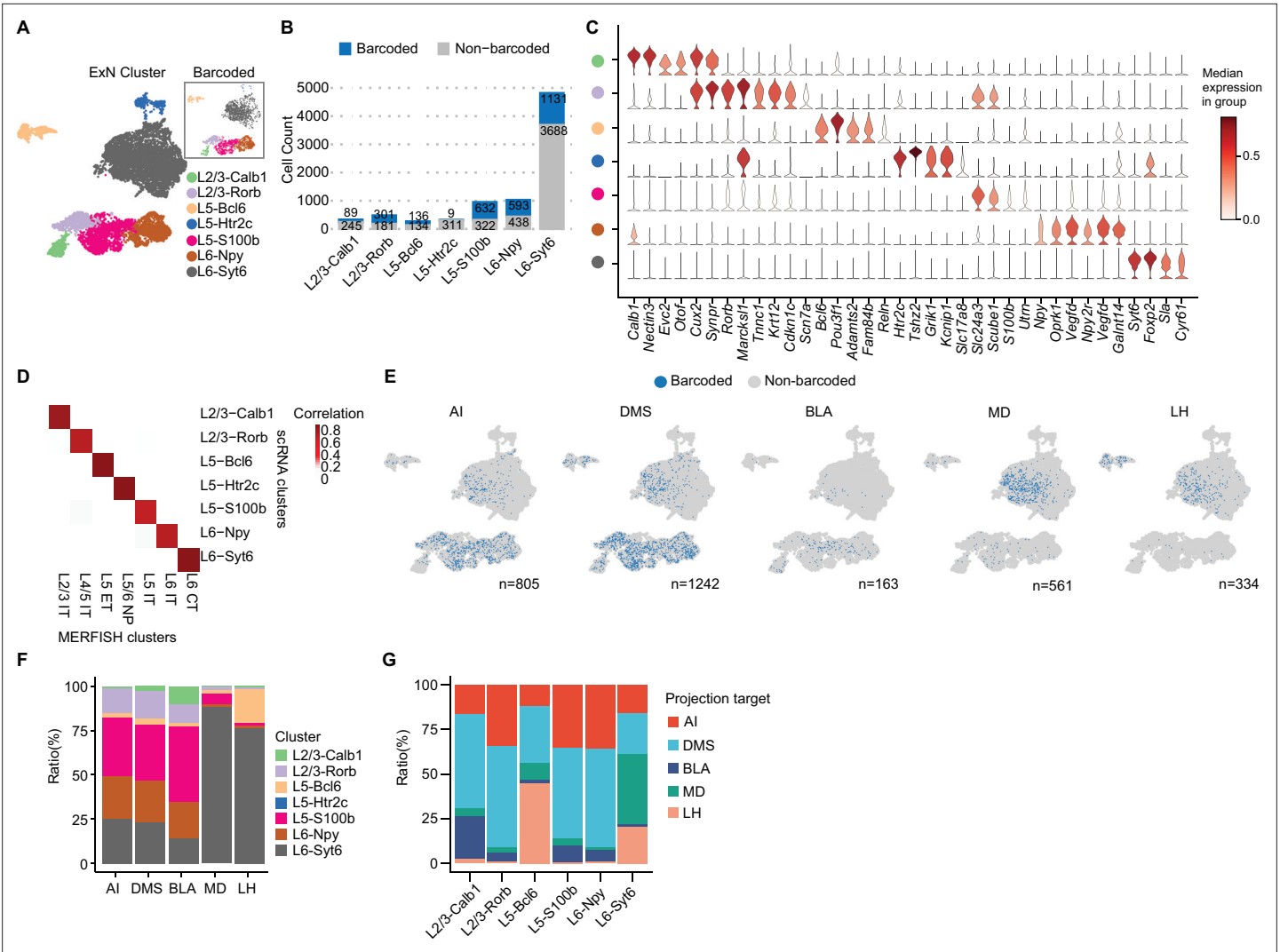

**Figure 2.** MERGE-seq unravels transcriptomic heterogeneity of projection target-defined vmPFC neurons. (**A**) Umap embedding of excitatory neuron subtype annotation. (**B**) Bar plot showing frequency of barcoded (blue) and non-barcoded (grey) neurons in distinct neuron subtypes. (**C**) Stacked violin plot showing the expression of markers for each neuronal subtype. (**D**) Heatmap showing the gene-expression correlation between excitatory subtypes defined by Multiplexed Error-Robust Fluorescence in situ Hybridization (MERFISH) and scRNA-seq. MERFISH data were downloaded from *Bhattacherjee et al., 2023*. (**E**) Umap embeddings of barcoded (blue) neurons projecting to each target. Number indicates the number of barcoded cells for each target. (**F**) Bar plot describing the distribution of neuronal subtypes for barcoded neurons associated with each projection target. Neuronal subtype color codes are the same as in (**A**), number of barcoded cells are same as the number indicated in (**E**) for each target. (**G**) Bar plot describing the distribution of projection targets for barcoded neurons associated with each neuronal type. In (**A, C, D**), 9368 cells in total were represented. In (**B, E, F**), 8210 cells in total were represented. In (**G**), cell numbers represented are as follows: L2/3-Calb1=72 cells, L2/3-Rorb=331 cells, L5-Bcl6=145 cells, L5-S100b=766 cells, L6-Npy=526 cells, L6-Syt6=1264 cells.

The online version of this article includes the following figure supplement(s) for figure 2:

**Figure supplement 1.** Layer and cluster annotation using the mouse brain atlas and published scRNA-seq transcriptomes, and projection patterns per mouse.

multiple targets. For instance, L5-S100b, L6-Npy and L2/3-Rorb mainly projected to AI, DMS and BLA, while L6-Syt6 mainly projected to MD and LH (*Figure 2G*). Interestingly, we also found that different neuron subtypes localized in the same layer could project to distinct targets. For instance, L6-Npy neurons projecting to AI, DMS, and BLA, while L6-Syt6 neurons projecting to MD, DMS, and LH (*Figure 2G*). Similar phenotypes were observed for L5-S100b and L5-Bcl6 subtypes (*Figure 2G*), suggesting transcriptomic and projection/function diversities in the spatially close neurons within the same cortical layer.

Together, by MERGE-seq analysis, we have revealed the heterogeneity and cellular composition of vmPFC neurons projecting to different target. Our results demonstrate that vmPFC neurons projecting to a certain target are composed of different transcriptome-defined neuronal subtypes, and individual transcriptome-defined subtypes of vmPFC neuron project to multiple targets.

## MERGE-seq reveals dedicated and collateral projection patterns of vmPFC neuron at single cell level

Interestingly, we found that a portion of barcoded vmPFC neurons had more than one type of barcode, suggesting collateral projection of these neurons. We therefore analyzed the projection pattern of each barcoded vmPFC neuron by calculating the number of valid barcode types (see Materials and methods). We defined the dedicated projection neuron as a neuron containing only one type of barcode, the collateral projection neuron as a neuron containing more than one type of barcode. We found 64.88% of 2050 viral-barcoded neurons belonged to dedicated projection and the remaining belonged to collateral projection. A total of 23.37% had dual targets (bifurcated projection), 8.15% had triple targets, and 3.61%, if any, projected to more than three targets (*Figure 3A*). It is worth mentioning that the definition of 'dedicated' and 'collateral' projections relies solely on the analysis of MERGE-seq data. The quantitative resolution of dedicated and collateral projections of vmPFC neurons will depend on the comprehensiveness of retrograde labeling from all postsynaptic targets and labeling efficiency. By calculating the conditional probability that the same neuron projects to two targets (see Materials and methods), we found that vmPFC neurons projecting to AI or BLA were more likely to have collateral projection to DMS (*Figure 3B*). We also observed a relatively high conditional probability of collateral projection between MD and LH, or DMS and LH, or DMS and MD (*Figure 3B*), suggesting bifurcated projections to these paired targets for single vmPFC neuron.

We first validate the bifurcated projection patterns (2 targets) inferred from the digital projectome. We injected retrograde AAV2 encoding different fluorescent proteins (EGFP or tdTomato) into different combinations of projection targets (dual-color rAAV2-retro labeling assay), and analyzed the projection patterns by immunohistochemistry. Consistent with MERGE-seq identifying DMS + LH bifurcated projections (*Figure 3B*), dual-color labeling revealed 17.8% ± 0.11% of vmPFC neurons collateralize to DMS and LH (*Figure 3—figure supplement 1A–C*). Of these, 73.28% ± 7.60% localized to layer 5 (*Figure 3—figure supplement 1A–C*). Other bifurcated projection patterns inferred by MERGE-seq was also verified by our dual-color retro-AAV labeling assay. These patterns included DMS + AI (23.1% ± 2.03% of all dual-color neurons) and DMS + BLA (6.59% ± 1.55%) (*Figure 3—figure supplement 1D–I*). In contrast, we only observed 1.66% ± 0.92% of dual-color labeled neurons in BLA + LH group (*Figure 3—figure supplement 1J–L*). This result is consistent with our MERGE-seq analysis, in which BLA + LH was not inferred as bifurcated projection targets (*Figure 3B*), further supporting the accuracy of the digital projectome based on MERGE-seq analysis.

Since dual-color labeling can only validate two targets, we additionally validated inferred projections by quantifying MERGE-seq patterns as percentages of totals and comparing to published single-neuron PFC projectome data (*Gao et al., 2022*). We found that DMS, AI + DMS, MD, and LH projection pattern appear as the most frequent projection patterns in both studies, with a relatively higher percentage of DMS dedicated projection pattern in MERGE-seq data (*Figure 3C*). We further categorized projection patterns by number of targets and found no significant differences versus imaging-based reconstruction (*Figure 3D*), indicating MERGE-seq faithfully identifies projection patterns.

Next, we focused our analysis on the 5 dedicated projections (DMS, AI, MD, LH, and BLA) and most frequent five collateral projections (DMS + AI, DMS + MD, DMS + LH, DMS + AI + MD, and DMS + AI + MD + LH). We conducted a principal component analysis (PCA) of the projection matrix and mapped binary projection labels on PC embeddings. Results from binary projection clustering aligned well with clusters at PC1- and PC2-defined embeddings (*Figure 3—figure supplement 1M*). We further clustered cells according to projection strength (defined as normalized projection barcode UMI counts; *Figure 3E*). We found that cells exhibited collateral projections to DMS + AI, or DMS + MD, or DMS + LH, or DMS + AI + MD, or DMS + AI + MD + LH (*Figure 3E*), a pattern very similar to that we observed in binary projection model, indicating that projection strength-based clustering is comparable to binary projection pattern model (*Figure 3B*). We next explored the cell type composition of the top 10 dedicated or collateral projection neurons. We mapped transcriptomic clusters

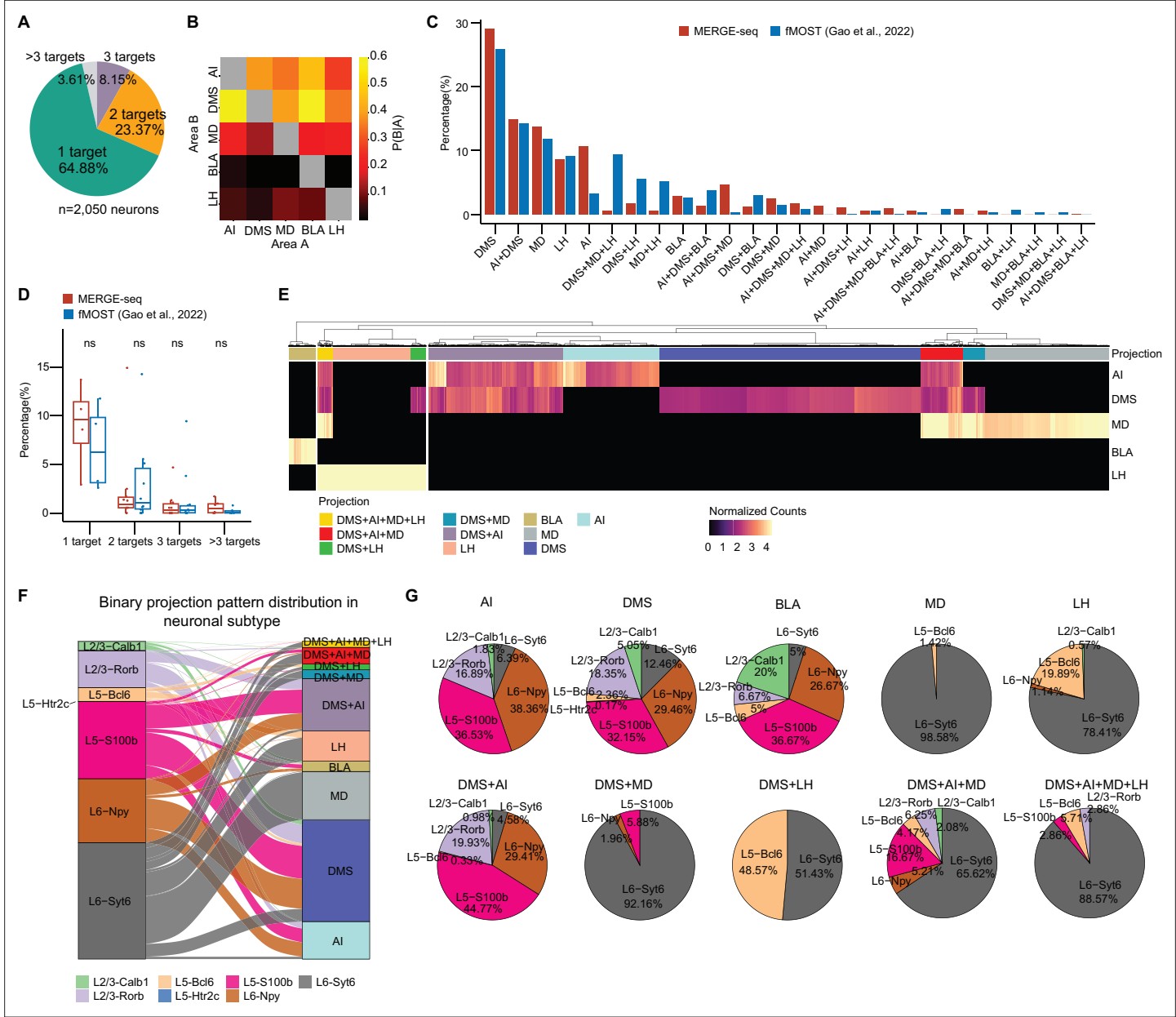

**Figure 3.** MERGE-seq reveals projection diversity within the vmPFC. (**A**) Pie chart indicating the number of projection targets for barcoded vmPFC neurons recovered by MERGE-seq. (**B**) Heatmap showing the probability that a neuron projecting to area A also projects to area B. (**C**) Bar graph illustrating the percentage of neuronal projection pattern of all projection patterns given five projection targets inferred by MERGE-seq (red bars) versus the 1155 fMOST-based single-neuron projectome data (blue bars) (**Gao et al., 2022**). (**D**) Boxplot comparison of percentage of neurons with different projection targets identified by MERGE-seq and fMOST. (**E**) Heatmap showing normalized projection strength. Rows represent the projection targets and columns represent the cells labeled by the top 10 binary projection patterns or labeled by transcriptional neuron subtypes. (**F**) Alluvial plot showing the 10 most frequent projection patterns distribution into neuronal subtypes. (**G**) Pie charts describing the projection patterns from (**E**) partitioned by neuronal subtype. In (**A, B**), 2050 barcoded neurons were represented. In (**C, D**), 2050 barcoded neurons from MERGE-seq data were represented, 1155 cells with fMOST data were represented (**Gao et al., 2022**). In (**E–G**), 1853 barcoded neurons (top 10 frequent projection patterns) were represented.

The online version of this article includes the following source data and figure supplement(s) for figure 3:

**Source data 1.** Related to *Figure 3C, D, F and G*.

**Figure supplement 1.** Immunostaining of dual-color, retrogradely labeled neurons and quantification, PCA plot of projection clusters.

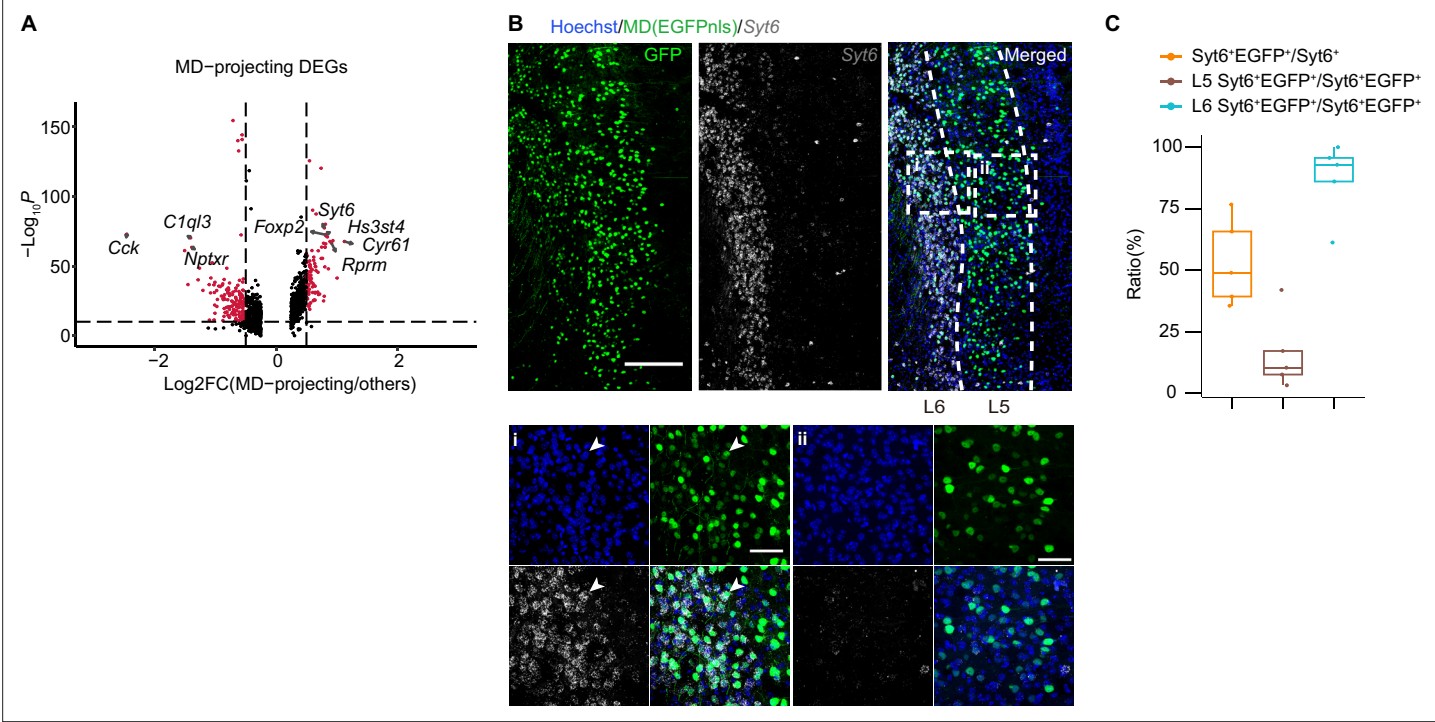

**Figure 4.** Transcriptional profiling of projection target-specific vmPFC neurons. (**A**) Volcano plots DEGs of MD-projecting versus non-MD-projecting vmPFC neurons. Assigned DEGs (red dots) were determined using threshold: $Log_2$ fold change = 0.5, p value cutoff=$10^{-10}$. (**B**) Immunostaining of EGFP (MD) and tdTomato (LH), and RNA FISH of *Syt6*. (i, ii) Enlarged view of dotted box in (**B**). (i) represents typical view at layer 6 and (ii) represents typical view at layer 5. Arrow head indicates *Syt6*+EGFP+ neurons. (**C**) Quantifications of (**B**). (**B**) Scale bars, 200 µm. i, ii in (**B**) Scale bars, 50 µm. N=3 mice. Data are presented as mean ± SD. In (**A**), 8210 cells were represented.

The online version of this article includes the following source data and figure supplement(s) for figure 4:

**Source data 1.** Related to *Figure 4A*, *Figure 4—figure supplement 1*.

**Source data 2.** Related to *Figure 4B and C*.

**Figure supplement 1.** Transcriptional profiling of projection target-specific vmPFC neurons.

to projection patterns (*Figure 3F*). While dedicated and collateral projection neurons were largely transcriptionally diverse (≥3 subtypes, *Figure 3G*), certain projections like MD-projecting and DMS + MD-projecting were highly homogeneous, composed of >90% L6-Syt6 cells (*Figure 3G*).

Overall, MERGE-seq elucidated dedicated and collateral vmPFC neuron projections at the single-neuron level, demonstrating diversity in projection patterns within individual vmPFC neurons. Furthermore, projection-defined (collateral or bifurcated) neurons have specific cell type composition and layer distributions. It is worth noting that as a proof of concept, we only acquired the vmPFC projectome from five downstream targets. Definitions of dedicated or collateral projections are thus limited to these five targets and some collateral projections may be underestimated.

## Transcriptional profiling of projection target-specific vmPFC neurons

Next, we sought to determine the molecular features of neurons projecting to different downstream targets. We calculated DEGs for each target-specific projection neurons (*Figure 4A*, *Figure 4—figure supplement 1*). We found that some of projection-specific DEGs are marker genes of typical neuronal types. For example, *Syt6*, *Foxp2*, and *Cyr61* are both MD-projecting DEGs and marker genes of L6-Syt6 neurons; *Rorb* and *Slc24a3* are both DMS-projecting DEGs and marker genes of layer 2/3 neurons (neuronal subtype L2/3-Rorb; *Figure 2C*, *Figure 4A*, *Figure 4—figure supplement 1*).

We further validated the molecular features of neurons associated with their specific projections by combining RNA fluorescence in situ hybridization (FISH) and retrograde labeling. *Syt6* is one of the DEGs of MD-projecting neurons (*Figure 4B*), and is the marker gene of L6-Syt6 cluster. By retrograde labeling of MD-projecting neurons and *Syt6* FISH experiment, we found that about 51.6% ± 16.9%

*Syt6*+ neurons project to MD. Further statistical analysis showed that, among *Syt6*+ MD-projecting (*Syt6*+EGFP+) neurons, 84.2% ± 14.8% were located in layer 6 while 15.8% ± 14.8% were located in layer 5 (*Figure 4C*), similar to the pattern obtained in our MERGE-seq analysis (*Figure 2F*). These results are in accordance with single-neuron projectomic and transcriptomic analysis of MERGE-seq, indicating that MERGE-seq can faithfully reveal the transcriptomic features of projection-specific neurons.

## MERGE-seq uncovers the molecular features of collateral projection neurons in vmPFC

Axons of projection neurons, including vmPFC neurons, have highly complex collaterals, which could regulate information processing and neural response properties at the microcircuit level (*Gagnon and Parent, 2014*; *Gao et al., 2022*; *Rockland, 2019*). However, the molecular features of neurons sending collateral projections remain elusive. MERGE-seq provides an opportunity to explore. Here, we identified DEGs for neurons with dedicated and collateral projection pattern (*Figure 5A*). Next, we asked whether there was transcriptional difference between neurons with dedicated projection to A and neurons with bifurcated projection to A and B. DEGs were rare in comparisons between projection patterns A/B vs. A, or A/B vs. B in all of groups we tested, except for the DMS + LH group and DMS + MD group (*Figure 5B and C,Figure 5—figure supplement 1*). We found that DMS + LH projection neurons were transcriptionally distinct to DMS but similar to LH, and DMS + MD neurons were transcriptionally distinct to DMS but similar to MD (*Figure 5B and C,Figure 5—figure supplement 1*). Specifically, we identified a set of genes which differentially expressed in DMS + LH projection neurons (such as *Pou3f1*, *Igfbp4,* and *Gprc5b*) or DMS + MD projection neurons (such as *Rprm*, *Crym*, *Hs3st4* and *Bc1*). Interestingly, *Pou3f1* is marker gene of L5-Bcl6 neurons (layer 5 neuron subtype), representing one of the two distinct neuron subtypes within the DMS + LH projection neuronal population (*Figure 3G*). We next verified the specific gene expression in DMS + LH projection neurons by using RNA FISH in combination with dual-color retrovirus labeling assay (*Figure 5D*). We found that the expression of *Pou3f1* was mainly distributed in layer 5, where *Pou3f1* was specifically expressed in dual-color labeled DMS + LH projecting neurons (white arrowheads, *Figure 5E*) and LH projecting neurons (white arrows, *Figure 5E*), but not DMS projecting neurons (blue arrows, *Figure 5E*). Quantification analysis showed that, among *Pou3f1*+ neurons, there are 55.7% ± 10.4% DMS + LH-projecting (*Pou3f1*+EGFP+tdT+) neurons, 31.6% ± 13.1% dedicated LH-projecting (*Pou3f1*+EGFP-tdT+) neurons, and 8.89% ± 2.38% dedicated DMS-projecting (*Pou3f1*+EGFP+tdT-) neurons (*Figure 5G*). We additionally discovered that 3.79% ± 2.91% of *Pou3f1*+ neurons did not project to either DMS or LH (*Pou3f1*+EGFP-tdT-) (yellow arrowheads, *Figure 5F*, *Figure 5G*). These results are consistent with our observation based on MERGE-seq data (*Figure 3G*).

Together, by MERGE-seq analysis and experimental validation, we uncovered that *Pou3f1* predominantly marks neurons projecting to the LH, denoting a distinct subset with collateral projections to both DMS and LH.

## Machine learning-based modeling reveals gene clusters for predicting projection patterns

Although many efforts have been made to correlate gene expression with neuronal circuit connectivity (*Huang et al., 2020*; *Sorensen et al., 2015*; *Sun et al., 2021*), the lack of a shared coordinate system for two modalities or limited genes examined reduces the prediction precision. MERGE-seq overcomes these challenges by acquiring high-throughput gene expression and projection pattern in the same neuron (*Figure 6A*). To evaluate potential relationships between the transcriptome and projectome, we used a probabilistic classifier, Naïve Bayes classifier, to predict binary projection patterns for each projection target based on transcription profiles. First, we encoded binary projection labels for each target region, encompassing both barcoded and non-barcoded projections, and subsequently trained a separate set of models for each of the five targets: AI, DMS, BLA, LH, and MD (see Materials and methods). Subsequently, we conducted a systematic evaluation of the impact of varying numbers of highly variable genes (HVGs), ranging from 2 to 5000, on model performance. This analysis revealed that employing the top 50 HVGs for modeling yielded the the highest F1 score (a harmonic mean of precision and recall), area under the curve (AUC), and a comparatively high prediction accuracy (see Materials and methods, *Figure 6—figure supplement 1A*). Next, we chose top 50 HVGs as features

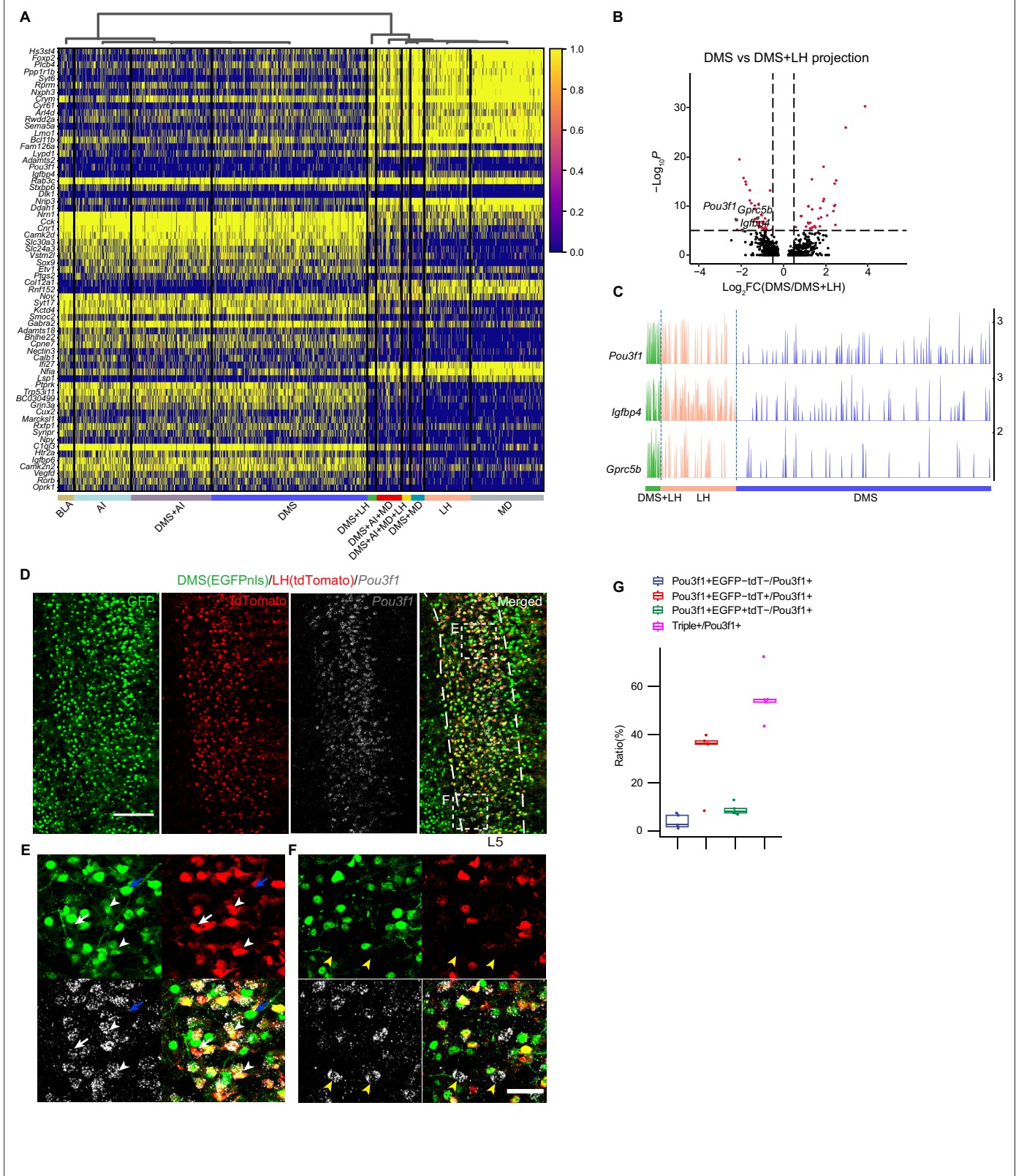

**Figure 5.** Molecular features of single vmPFC neuron with collateral projections to downstream targets. (**A**) Heatmap showing scaled expression of calculated DEGs based on 10 projection patterns. Top 10 DEGs ordered by average $\log_2$ fold change of each pattern were selected. (**B**) Volcano plot showing genes differentially expressed in the DMS + LH-bifurcated projection pattern compared to the DMS-dedicated projection pattern. (**C**) Track plots showing normalized data of the selected DEGs in DMS-dedicated, LH-dedicated, and DMS + LH-bifurcated projection pattern. (**D–F**) Examining

*Figure 5 continued on next page*

*Figure 5 continued*

*Pou3f1* and DMS + LH-bifurcated projection pattern using RNA FISH and immunostaining of dual-color traced retrograde labeled neurons. Virus injection scheme was the same as in *Figure 3—figure supplement 1*. Scale bars, 200 μm. (**E, F**) Enlarged view of dotted box in (**D**). Arrow heads indicate *Pou3f1*+EGFP+tdTomato+ neurons, white arrows indicate *Pou3f1*+EGFP-tdTomato+ neurons, blue arrows indicate *Pou3f1*-EGFP+tdTomato- neurons, and yellow arrowheads indicate *Pou3f1*+EGFP-tdTomato- neurons. Scale bars, 50 μm. (**G**) Quantification of (**D**). N=3 mice, Data are presented as mean ± SD. In (**A**), 1,853 barcoded neurons (top 10 frequent projection patterns) were represented. In (**C**), 805 barcoded neurons (projection pattern DMS + LH = 35, LH = 176, DMS = 594) were represented.

The online version of this article includes the following source data and figure supplement(s) for figure 5:

**Source data 1.** Related to *Figure 5B*.

**Source data 2.** Related to *Figure 5D–G*.

**Figure supplement 1.** DEGs between dedicated projection neurons versus bifurcated neurons.

to build the model. As a control model, we chose 50 randomly chosen genes. Five projection targets models were independently trained by splitting cells into training (70%) and test dataset (30%). Using top 50 HVGs also gave rise to significantly better model performance in regarding to prediction accuracy, AUC and F1 score, compared to using randomly chosen 50 genes (*Figure 6B*). We also performed 100 iterations randomly sampling 1000 cells and swapping barcoded with non-barcoded labels, which substantially decreased model predictive performance across various evaluation metrics (see Materials and methods, *Figure 6—figure supplement 1B*). This outcome underscores the critical importance of label accuracy for the predictive capabilities of the model, suggesting the authenticity of current barcoded cells labels despite potential false positives from stringent UMI thresholding. Altogether, these results suggest that the top 50 HVGs are more informative for predicting and decoding projection patterns.

To interpret the important genes contributing to a certain projection pattern, we used a game-theoretic approach to explain the output of HVGs-based Naïve Bayes models (*Lundberg et al., 2020*; *Figure 6A*). We used top 50 HVGs to build Naïve Bayes model and summarized effects of HVGs in SHAP (SHapley Additive exPlanations) values for each projection pattern (see Materials and methods; *Figure 6C–F*, *Figure 6—figure supplement 1C*). As examples, *Nptxr* gene was the top positive predictors for DMS projection, suggesting that a cell that expresses high levels of *Nptxr* has a higher probability of projecting to DMS. Similarly, *Rprm* was the top positive predictors for MD projection. By examining top effective genes (features) on PC embeddings of the projection matrix, we found that the expression pattern of these positive predictors mostly overlaps with projection barcode distribution (*Figure 6D and F*). These results mathematically establish the relationship between gene expression and structural connectivity, indicating the predictive power of a specific gene cluster for projection properties of vmPFC neurons.

## Discussion

Given the complexity of brain circuits, neuronal subtypes must be characterized from multiple viewpoints (*Zeng, 2022*). Information including neuronal projection patterns (i.e. region-to-region connectivity), physiological properties, gene expression, and how they encode information in behavioral paradigms, are essential to understand functional brain circuits. Therefore, it is inevitably difficult to acquire a complete picture of brain circuits when only one analytic modality is considered. In this study, we have developed a multiplexed barcoding method that is integrated with scRNA-seq, enabling simultaneous transcriptome and projectome analyses. Retrograde AAVs are injected into multiple target regions simultaneously, thereby labeling projection neurons within the brain region of interest and facilitating their transcriptional analysis. Here, by comparing to other methods, we highlight distinct features of MERGE-seq and key biological insights that MERGE-seq can provide.

Early approaches of barcode-based neuronal projection mapping mainly focus on elucidating the projections of individual neurons in a single brain without providing the transcriptional signatures corresponding to those individual neurons (MAPseq; *Kebschull et al., 2016*). We therefore developed MERGE-seq to connect single-neuron transcriptome and projectome with high throughput. While there are some conceptual similarities to BARseq or ConnectID (*Chen et al., 2019*; *Klingler et al., 2021*), MERGE-seq has its unique features and advantages. BARseq can acquire single-neuron transcriptome and projectome, but with only a number of genes due to limited throughout of in situ

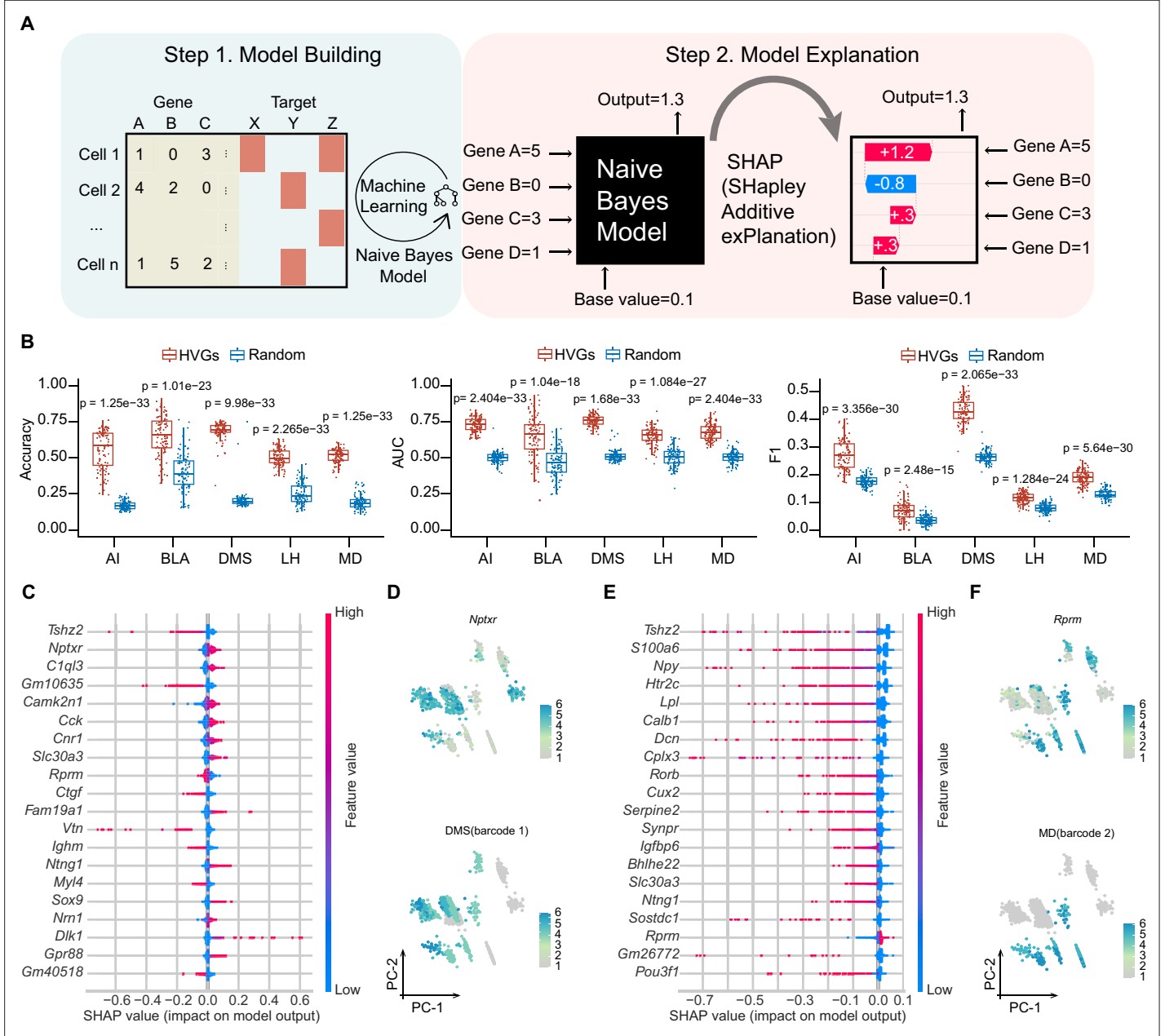

**Figure 6.** Machine learning-based modeling predicts projection patterns based on gene expression. (**A**) Schematics of machine learning modeling steps. (**B**) Prediction accuracy (left panel), AUC score (middle panel) and F1 score (right panel) of top HVGs and random chosen equal number of genes for modeling building. A total of 100 trials have been performed by randomly sampling 1000 cells from 8210 cells. Top 50 HVGs or 50 randomly chosen genes were used as features per trial. Comparisons were made between models built by the HVGs group and random genes group for each projection target. The displayed p value was computed using a two-sided Wilcoxon test. Data are the mean ± SD. (**C, E**) SHAP summary plots of DMS and MD showing important features (genes) with feature effects. For each model, non-barcoded cells were encoded to class 0 and barcoded cells were encoded to class 1. Models were built using top 50 HVGs. (**D, F**) Normalized expression of the most important genes with positive feature effects in Naïve Bayes modeling of DMS (**D**) or MD (**F**) and normalized expression of barcode 1 representing DMS-projecting (**E**) or barcode 2 representing MD-projecting (**F**) on PC1 and PC2 embeddings. Note that bottom panel of (**D, F**) is identical to DMS and MD barcode expression in *Figure 3—figure supplement 1M*. In (**D, F**), 1853 barcoded neurons (top 10 frequent projection patterns) were represented. In (**C, E**), For calculating SHAP values, both the training and testing datasets were subsampled to include 1500 cells each.

The online version of this article includes the following source data and figure supplement(s) for figure 6:

**Source data 1.** Related to *Figure 6B*, *Figure 6—figure supplement 1B*.

**Figure supplement 1.** SHAP summary plots of Naïve Bayes models.

sequencing. An improved version of BARseq can allow tens of genes to be detected, but still with a low throughput compared to scRNA-seq and a high cost in regards to synthesizing RNA probes (*Sun et al., 2021*). ConnectID (scRNA-seq combined with MAPseq) improves the detection of transcriptome using scRNA-seq but has a relatively low recovery rate of cells with transcriptome and projectome simultaneously (~16%, 391 cells with barcode identity in 2450 cells with scRNA-seq; *Klingler et al., 2021*). In contrast, MERGE-seq enables transcriptional profiling of thousands of genes per neuron, with valid projectome barcode information recovered from approximately 50% of FAC-sorted cells passing stringent determination criteria. Another advantage of MERGE-seq is that users only need to sequence one brain region – the source area. While, in BARseq or ConnectID, users need to perform numerous tissue homogenization and sequencing for downstream brain regions to query target area barcodes information (projection).

MERGE-seq is a retro-AAV-based scRNA-seq approach. Previous research has employed retro-AAV techniques to probe the projection-specific transcriptome or epigenome of individual neurons (*Lui et al., 2021*; *Tasic et al., 2018*; *Tasic et al., 2016*; *Yao et al., 2021*; *Zhang et al., 2021*). Yet, these studies have not developed a multiplexed approach for investigating the complex collateral projection patterns of neurons. Another retro-AAV based-approach, VECTORseq, was recently developed to associate neuronal projectome and transcriptome (*Cheung et al., 2021*). VECTORseq used several viral transgenes including three recombinases (DreO, Cre, Flpo) and two fluorescent proteins (tdTomato and EGFP) to barcode neurons. However, these transgenes are variable in length (DreO, ~1000 bp; Cre, ~1000 bp; FLPo, ~1200 bp; tdTomato, ~1400 bp; EGFP, ~700 bp) and driven by different promoters with different strength (EF1a, hSyn, CAG). Such an approach will inevitably result in differential expression of these different transgenes in labelled neurons, which in turn leads to different rates of transgene recovery in these neurons. In addition, viral-mediated overexpression of these recombinase may lead to toxic to the labeled neurons due to non-specific recombination events (*Xiao et al., 2012*). Therefore, the transgenes used in VECTORseq method should be carefully selected to avoid any potential interferences with neuronal function or gene expression by these different transgenes. In contrast, MERGE-seq used 15-nucleotide barcode sequences in the 3'UTR region of EGFP as projection index driven by the same promoter to label different projection neurons. The expression of these different barcoded EGFP mRNA is comparable, and the number of these barcoded retro-AAV is unlimited. Therefore, MERGE-seq allows users to examine more populations (theoretically unlimited) in one brain and more extensive analysis of collateralization. Further, MERGE-seq can reveal projectome of single collateral projection neurons and identify molecular features of these neurons (*Figure 5*). However, the collateral projection patterns of single neurons were not reported in VECTORseq and Retro-seq-based method (*Cheung et al., 2021*; *Lui et al., 2021*). For example, *Lui et al., 2021*. used Retro-seq to investigate the correspondence between transcriptomics and projection patterns of vmPFC neurons, and inferred collateral projection based on the finding that transcriptome-defined neuron subtypes can project to different targets (or neurons projecting to different targets share common transcriptome-defined neuron subtype). However, the population level multi-target projections of a transcriptome-defined neuron subtype do not necessarily reflect collateral projection of individual neurons within a subtype. For instance, individual neurons within a subtype could project to distinct targets (dedicated projection), but their collective projections show multiple targets. In contrast, in MERGE-seq, individual neurons that were retrogradely labeled multiple projection barcodes are determined as collateral projection neurons. By MERGE-seq analysis, we uncovered dedicated and collateral projection patterns of individual vmPFC neurons to the five downstream targets, and revealed molecular features associated with these dedicated or collateral projection neurons (*Figures 3–5*). In addition, MERGE-seq strategy can be readily applied to other animal models, which is especially beneficial for research in models (e.g. non-human primate) where genetic manipulation is challenging. In summary, while Retro-seq methods provide valuable population-level insights, they do not capture the complex collateral projections that MERGE-seq can discern at the single-cell level. Our findings build upon and extend those of Lui et al. by demonstrating that individual neurons within transcriptome-defined subtypes exhibit a diverse range of projection patterns. This contributes a new layer of understanding to the intricate architecture of PFC circuits, emphasizing the nuanced interplay between divergence and convergence in neuronal pathways.

Although MERGE-seq does not offer spatial information of neurons currently, it leverages widely accessible droplet-based scRNA-seq, avoiding specialized equipment. Meanwhile, the extensive

spatially resolved mouse brain atlases available (*Allen et al., 2023*; *Yao et al., 2023*; *Zhang et al., 2023*) allow for easy spatial annotation of cell populations using DEGs identified by scRNA-seq, as we demonstrated by mapping neuronal subtypes with MERFISH data of PFC. Compared to imaging-based spatial transcriptomics like MERSCOPE with constrained gene numbers, or next-generation sequencing (NGS)-based methods that are lack of true single-cell resolution (e.g. 50 µm 10 x Genomics Visium or 10–50 µm for DBiT-seq-based methods; *Deng et al., 2023*), we believe our method stands out as a robust solution and offers an advantageous balance between resolution and scope.

There are several potential concerns and limitations of current study. First, a recognized limitation of using retro-AAV-based methods, including MERGE-seq, is the imperfect retrograde labeling efficiency in target regions. Labeling efficiency could be variable depending on the different source brain regions, projection strength, the distance between source and target brain regions and different AAV serotypes or tropism. For example, only nine neurons of the L5-Htr2c subtype were recovered with valid barcodes, which may be attributable to technical factors including cell loss during dissociation or AAV2-retro tropism. Alternatively, this subtype may intrinsically lack projections to the selected target regions examined in this study. Furthermore, single-cell dissociation for scRNA-seq can result in cell loss, thereby reducing the recovery rate of barcoded neurons. All these factors could influence the extent to which the complete range of neuronal projections is captured. Consequently, the quantitative conclusions drawn here might not fully represent the true extent of neuronal projections.

Second, the robust detection of projection barcodes and its recovery rate in neurons labeled with barcoded AAV-retro viruses is indeed a critical and challenging aspect of our methodology. As mentioned above, this challenge is largely due to the differential viral transduction efficiency across neurons, leading to inconsistent barcode expression. Neurons with low barcode expression may fall beneath the detection threshold of conventional sequencing methods. A suboptimal recovery rate can potentially lead to underrepresentation of certain neuron populations or projection patterns in the analyzed data. This in turn could impact the interpretation of neuronal connectivity and function, as projections that are less efficiently labeled or harder to detect might be overlooked. For instance, if a subset of neurons with low barcode expression is systematically missed, it could erroneously suggest that these neurons do not participate in specific projection patterns. Conversely, overrepresentation of certain barcodes due to higher transduction efficiency could falsely indicate a predominance of certain projections. One potential solution to improve barcode detection is to include FAC-sorted EGFP-negative cells as a negative control, which may help to differentiate between true signal and background noise. Enhancements in sequencing technologies, offering increased read lengths and deeper sequencing, could potentially improve barcode detection sensitivity. In parallel, applying single-molecule FISH technologies like MERFISH to spatially resolve barcodes offers a robust and direct detection method. This technology can provide detailed coverage and resolution of individual RNA molecules within single cells, bypassing additional PCR amplification steps and reducing cell loss during physical isolation. Furthermore, carefully controlling the viral titer and refining the procedures of single neuron suspension preparation, as performed in this study, is required to control the labeling efficiency and recovery rate.

DMS is en route from vmPFC to subcortical regions (*Shepherd, 2013*), thus raising another concern about the transducing ability of AAV2 in axons of passages. However, the retrograde transport of AAV has been effectively demonstrated to target projection neurons at axonal terminals, with injections into the DMS exhibiting labeling patterns and efficiencies that match those of synthetic tracers (*Tervo et al., 2016*). Further, it has been experimentally verified that AAV2 spread is confined to the vicinity of synaptic terminals and does not affect axon fibers in passages, especially as evidenced by retro-AAV injections in the cervical spinal cord (*Wang et al., 2018*). While these findings are reassuring, additional research is needed to unequivocally eliminate the possibility of transduction along axon fibers of passage. The five distinct injection sites we chose for our study are spatially disparate, encompassing both cortical and subcortical regions, and span a range from the anterior (Bregma,+2 mm) to the posterior (Bregma, –1.5 mm) brain regions. This separation mitigates the potential overlap in labeling when examining spatially proximate nuclei, such as those in the hypothalamus. Nevertheless, examining such closely situated targets would necessitate meticulous quantification of virus injection volumes to prevent cross-target viral dissemination, ensuring the specificity required for accurate projection mapping.

In summary, we develop MERGE-seq, a powerful multiplexed projectome and transcriptome analysis platform that will help researchers perform big-data research at low cost. This will enable researchers to understand organizing principles and molecular features of neural circuits across modalities, and to construct more comprehensive mesoscale connectomes.

# Materials and methods

**Key resources table**

| Reagent type (species) or resource | Designation | Source or reference | Identifiers | Additional information |
|---|---|---|---|---|
| Antibody | Anti-GFP (Rat Monoclonal) | Nacalai | Cat# 04404–84, RRID:AB_10013361 | IHC(1:500) |
| Antibody | Anti-tdTomato (Goat Polyclonal) | OriGene | Cat#AB8181-200, RRID:AB_2722750 | IHC(1:500) |
| Antibody | Hoechst 33342 | Lifetech | Cat#H3570 | IHC(1:1000) |
| Antibody | Donkey anti-rat Alexa Fluor 488 (Donkey Polyclonal) | Invitrogen | Cat#A21208 | IHC(1:800) |
| Antibody | Donkey anti-goat Alexa Fluor 568 (Donkey Polyclonal) | Invitrogen | Cat#A11057 | IHC(1:800) |
| Recombinant DNA reagent | pAAV-CAG-tdTomato (plasmid) | Addgene | Cat#59462 | |
| Recombinant DNA reagent | pAAV-CAG-EGFP barcode-0-SV40 polyA (plasmid) | This paper | Cat#190864 | Submitted to Addgene |
| Recombinant DNA reagent | pAAV-CAG-EGFP barcode-1-SV40 polyA (plasmid) | This paper | Cat#190865 | Submitted to Addgene |
| Recombinant DNA reagent | pAAV-CAG-EGFP barcode-2-SV40 polyA (plasmid) | This paper | Cat#190866 | Submitted to Addgene |
| Recombinant DNA reagent | pAAV-CAG-EGFP barcode-3-SV40 polyA (plasmid) | This paper | Cat#190867 | Submitted to Addgene |
| Recombinant DNA reagent | pAAV-CAG-EGFP barcode-4-SV40 polyA (plasmid) | This paper | Cat#190868 | Submitted to Addgene |
| Recombinant DNA reagent | pAAV-CAG-EGFP barcode-5-SV40 polyA (plasmid) | This paper | Cat#190869 | Submitted to Addgene |
| Recombinant DNA reagent | pAAV-CAG-EGFP barcode-6-SV40 polyA (plasmid) | This paper | Cat#190870 | Submitted to Addgene |
| Recombinant DNA reagent | pAAV-CAG-EGFP barcode-7-SV40 polyA (plasmid) | This paper | Cat#190871 | Submitted to Addgene |
| Recombinant DNA reagent | pAAV-CAG-EGFP barcode-8-SV40 polyA (plasmid) | This paper | Cat#190872 | Submitted to Addgene |
| Recombinant DNA reagent | pAAV-CAG-EGFP barcode-9-SV40 polyA (plasmid) | This paper | Cat#190873 | Submitted to Addgene |

*Continued on next page*

*Continued*

| Reagent type (species) or resource | Designation | Source or reference | Identifiers | Additional information |
|---|---|---|---|---|
| Recombinant DNA reagent | pAAV-CAG-EGFP barcode-10-SV40 polyA (plasmid) | This paper | Cat#190874 | Submitted to Addgene |
| Recombinant DNA reagent | pAAV-CAG-EGFPnls barcode-206-SV40 polyA (plasmid) | This paper | Cat#190875 | Submitted to Addgene |
| Recombinant DNA reagent | pAAV-CAG-EGFPnls barcode-210-SV40 polyA (plasmid) | This paper | Cat#190876 | Submitted to Addgene |
| Chemical compound, drug | AMPA receptor antagonist CNQX | Abcam | Cat#ab120017 | working concentration:10 µM |
| Chemical compound, drug | NMDA receptor antagonist D-AP5 | Abcam | Cat#ab120003 | working concentration:50 µM |
| Chemical compound, drug | 2-Mercaptoethanol | Sigma | Cat#M6250 | working concentration:0.067 mM |
| Chemical compound, drug | EDTA | Invitrogen | Cat#15575020 | working concentration:1.1 mM |
| Chemical compound, drug | L-Cysteine hydrochloride monohydrate | Sigma | Cat#C7880 | working concentration:5.5 mM |
| Chemical compound, drug | Deoxyribonuclease I | Sigma | Cat#D4527 | working concentration:100 units/ml |
| Chemical compound, drug | Protease | Sigma | Cat#P5147 | working concentration:1 mg/ml |
| Chemical compound, drug | Dispase | Worthington | Cat#LS02106 | working concentration:1 mg/ml |
| Chemical compound, drug | Papain | Worthington | Cat#LS003126 | working concentration:20 units/ml |
| Commercial assay or kit | Debris Removal Solution | Miltenyi | Cat#130-109-398 | |
| Commercial assay or kit | Chromium Single Cell 3' Reagent Kits (v3) | 10 X Genomics | Cat#PN1000075 | |
| Commercial assay or kit | NEBNext Ultra II Q5 Master Mix | NEB | Cat#M0544L | |
| Commercial assay or kit | SPRIselect | Beckman | Cat#B23317 | |
| Commercial assay or kit | Mm-Syt6 | ACD Bioscience | Cat#449641 | |
| Commercial assay or kit | Mm-Pou3f1-C2 | ACD Bioscience | Cat#436421-C2 | |
| Sequence-based reagent | P5-Read1 | This paper | PCR primers | AATGATACGGCGACCACCGAGATC TACACTCTTTCCCTACACGACGCTC |
| Sequence-based reagent | P7-index-Read2-EGFP | This paper | PCR primers | CAAGCAGAAGACGGCATACGAGATAGGATTCGG TGACTGGAGTTCAGACGTGTGCTCTTCCGATCT GgCATGGACGAGCTGTACAAG |

## AAV vector design

Plasmid pAAV-CAG-tdTomato (Addgene, #59462) was first modified by replacing tdTomato and WPRE with EGFP by T4 DNA Ligase mediated ligation. A 15 bp barcode sequence was then inserted after the stop codon of EGFP, linked by EcoRI restriction enzyme recognition site. Sequences barcode 0 representing the AI target, *CTGCACCGACGCATT*; barcode 1 (DMS target), *GAAGGCACAGAC TTT*; barcode 2 (MD target), *GTTGGCTGCAATCCA*; barcode 3 (BLA target), *AAGACGCCGTCG*

*CAA*; barcode 4 (LH target), *TATTCGGAGGACGAC*. Other barcode sequences used for IHC include barcode 10, *AGCTATGCACGATCA*; barcode 206, *GCGTAAGTCTCCTTG*; barcode 210, *CCTGTATG CGTGGAG*. Engineered viruses were produced by Gene Editing Core Facility, Center for Excellence in Brain Science and Intelligence Technology.

## Virus injection

Male adult C57BL/6 mice (8 weeks of age) were anesthetized intraperitoneally using pentobarbital sodium (10 mg/mL, 120 mg/kg b.w.) and unilaterally injected with rAAV2-retro-EGFP-Barcode virus (barcode 0, 1, 2, 3, 4) into five projection targets simultaneously. Coordinates for these injections are as follows. Reference from Bregma and dura, AI at two locations (in mm: 2.0 AP, 2.52 ML, –2.0 DV; 1.6 AP, 2.97 ML, –2.2 DV) with rAAV2-retro-EGFP-barcode 0 (250 nl and 200 nl, $2.90 \times 10^{13}$ VG/ml); DMS at one location (in mm: 0.6 AP, 1.8 ML, –2.2 DV, 8 degree angle), with rAAV2-retro-EGFP-barcode 1 (500 nl, $1.00 \times 10^{13}$ VG/ml); MD at one location (in mm: –1.25 AP, 1.35 ML, –3.55 DV, 20 degree angle), with rAAV2-retro-EGFP-barcode 2 (300 nl, $1.27 \times 10^{13}$ VG/ml); BLA at one location (in mm: –1.5 AP, 3.2 ML, –4.2 DV), with rAAV2-retro-EGFP-barcode 3 (300 nl, $2.00 \times 10^{13}$ VG/ml); LH at one location (in mm: –0.94 AP, 1.2 ML, –4.55 DV), with rAAV2-retro-EGFP-barcode 4 (250 nl, $2.25 \times 10^{13}$ VG/ml). Following each injection, the micropipette was left in the tissue for 10 min before being slowly withdrawn to prevent virus spilling and backflow. Mice were sacrificed 6 weeks after virus injection. Single-cell suspensions were generated as described in methods below.

For dual-color retrograde virus tracing, two regions were ipsilaterally injected with virus at the same time, one with rAAV2-retro-EGFP-barcode 10 ($2.00 \times 10^{13}$ VG/ml) or rAAV2-retro-EGFPnls-barcode 206 or 210 ($3.10 \times 10^{13}$ VG/ml for barcode 206 and $4.38 \times 10^{13}$ VG/ml for barcode 210) and one with rAAV2-retro-tdTomato ($2.25 \times 10^{13}$ VG/ml). rAAV2-retro-EGFPnls was used to avoid dense fiber staining when performing immunohistochemistry. We deposited the virus plasmid constructs to Addgene (pAAV-CAG-EGFP barcode-(0–10)-SV40 polyA, pAAV-CAG-EGFPnls barcode-(206, 210)-SV40 polyA; Addgene ID 190864–190876).

## scRNA-seq sample and library preparation

For mice without FAC-sorting (mouse #1, #2, #3), three mice that had been injected with virus were anaesthetized and then subjected to transcranial perfusion with ice-cold oxygenated self-made dissection buffer (in mM: 92 Choline chloride, 2.5 KCl, 1.2 $NaH_2PO_4$, 30 $NaHCO_3$, 20 HEPES, 25 Glucose, 5 Sodium ascorbate, 2 Thiourea, 3 Sodium pyruvate, 10 $MgSO_4.7H_2O$, 0.5 $CaCl_2.2H_2O$, 12 N-Acetyl-L-Cysteine). The brain was removed, 300 µm vibratome sections were collected, and the PrL and IL regions were microdissected under a stereo microscope with a cooled platform. Brain slices were incubated in dissection buffer with 10 µM AMPA receptor antagonist CNQX (Abcam, ab120017) and 50 µM NMDA receptor antagonist D-AP5 (Abcam, ab120003) at 33 °C for 30 min. The pieces were dissociated first using the ice-cold oxygenated dissection buffer added papain (20 units/ml, Worthington, LS003126), 0.067 mM 2-mercaptoethanol (Sigma, M6250), 1.1 mM EDTA (Invitrogen, 15575020), 5.5 mM L-Cysteine hydrochloride monohydrate (Sigma, C7880) and 100 units/ml Deoxyribonuclease I (Sigma, D4527), with 30–40 min enzymatic digestion at 37 °C, followed by 30 min 1 mg/ml protease (Sigma, P5147) and 1 mg/ml dispase (Worthington, LS02106) enzymatic digestion at 25 °C. Supernatant was removed and digestion was terminated using dissection buffer containing 2% fetal bovine serum (FBS, Bioind, 04-002-1A). Single-cell suspension was generated by manual trituration using fire-polishing Pasteur pipettes and filtered through a 35 µm DM-equilibrated cell strainer (Falcon, 352052). Cells were then pelleted at $400 \times g$ for 5 min. The supernatant was carefully removed and resuspended in 1–2 ml dissection buffer containing 2% FBS. The suspension was then subjected to the debris removal step using the Debris Removal Solution (Miltenyi, 130-109-398). Cell pellets were resuspended and 48,000 cells were loaded into 3 lanes to perform 10x Genomics sequencing. For mice with FAC-sorting (mouse #4, #5, #6), PrL and IL regions were microdissected and dissociated as mice without FAC-sorting, cells were sorted to enrich for EGFP-positive rAAV2-retro-EGFP-barcodes labeled cells. About 4893 EGFP-positive cells were captured and loaded to perform 10x Genomics sequencing. Chromium Single Cell 3' Reagent Kits (v3) were used for library preparation (10x Genomics). Libraries were sequenced on an Illumina Novaseq 6000 system.

## Projection barcode library preparation

Parallel PCR reactions were performed containing 50 ng of post cDNA amplification reaction cleanup material as a template. P5-Read1 (*AATGATACGGCGACCACCGAGATCTACACTCTTTCCCTACACGA CGCTC*) and P7-index-Read2-EGFP (*CAAGCAGAAGACGGCATACGAGATAGGATTCGGTGACTGG AGTTCAGACGTGTGCTCTTCCGATCTGgCATGGACGAGCTGTACAAG*) (200 nM each) were used as primers with the NEBNext Ultra II Q5 Master Mix (NEB, M0544L). Amplification was performed using the following PCR protocol: (1) 33 °C for 1 min, (2) 98° for 10 s, then 65 °C for 75 s (20–24 cycles), (3) 75 °C for 5 min. Reactions were re-pooled during 1 X SPRI selection (Beckman, B23317), which harvested virus projection barcodes library. 431–437 bp (with 120 bp adaptors) libraries were sequenced using Illumina HiSeq X Ten.

## Immunohistochemistry

Mice were sacrificed 6 weeks after virus injection. Mice were transcardially perfused with phosphate-buffered saline (PBS) followed by 4% paraformaldehyde (PFA). Brain samples were extracted and cryoprotected in 20% sucrose/4% PFA, immersed sequentially in 20% sucrose (in 4% PFA) and 30% sucrose (in 0.1 M phosphate buffer, PB) until sunk, and then transferred to 30% sucrose/PB for more than 24 h. Brain samples were flash-frozen on dry ice and sectioned at 30 μm on a cryostat (Leica, SM2010R). For dual-color retrograde virus tracing, brain slices were blocked in 10% donkey serum and 0.3% Triton X-100 at 37 °C for 1 hr. Slices were then incubated with primary antibodies against green fluorescent protein (GFP, 1:500, Nacalai, 04404–84, RRID: AB_10013361) and tdTomato (1:500, OriGene, AB8181-200, RRID: AB_2722750) at room temperature for 2 hr, then 4 °C overnight. Slices were washed three times using PBS and incubated with Hoechst 33342 (1:1000, Lifetech, H3570), as well as secondary donkey anti-rat Alexa Fluor 488 antibodies (1:800, Invitrogen, A21208) and donkey anti-goat Alexa Fluor 568 antibodies (1:800, Invitrogen, A11057) at room temperature for 1 hr. Slices were washed three times using PBS and coverslipped. Stained slices were imaged with a 4 X objective with numerical aperture 0.16 as a map, followed by 1.5 μm increment z stacks with a 10 X objective with numerical aperture 0.4 (FV3000, OLYMPUS). Composite images were automatically stitched in the X-Y plane using ImageJ/FIJI. RNA FISH experiments were performed using RNA-Scope reagents and protocols (ACD Bioscience, CA), following instructions for fixed-frozen tissue. For experiments using RNA-Scope, immunohistochemistry was performed following RNA-Scope. Probes of RNA-Scope used in this study include, Mm-Syt6 (449641), Mm-Pou3f1-C2 (436421-C2).

## scRNA-seq data pre-processing

scRNA-seq data were aligned with the customized mouse reference genome mm10-3.0.0 adding five projection barcodes as separate genes. Further projection barcode expression was obtained as described in (Projection barcode library preparation and Projection barcode FASTQ alignment). scRNA-seq data were demultiplexed using the default parameters of Cellranger software (10x Genomics, v3.0.2). Obtained filtered transcription count matrix was used for downstream analysis. For unsorted samples, we used three mice with three GEM wells in one Chromium Single Cell 3' Chip (v3). Among unsorted samples, sample mouse #1 recovered 8040 cells, 447,984,945 read pairs were aligned, mean reads per cell is 55,719, median genes per cell is 2382; sample mouse #2 recovered 7443 cells, 399,187,134 read pairs were aligned, mean reads per cell is 53,632, median genes per cell is 2379; sample mouse #3 recovered 7243 cells, 410,627,696 read pairs were aligned, mean reads per cell is 56,693, median genes per cell is 2385. For FAC-sorted samples, we used three mice with one GEM well in one Chromium Single Cell 3' Chip (v3). FAC-sorted sample recovered 2075 cells, 410,434,792 read pairs were aligned, mean reads per cell is 197,799, median genes per cell is 6533.

## Projection barcode FASTQ alignment

Demultiplexing of projection index barcode was performed using deMULTIplex R package (v1.0.2) (https://github.com/chris-mcginnis-ucsf/MULTI-seq, copy archived at *mcGinnis, 2023*) with modifications. Briefly, we have revised the MULTIseq.align function to count the UMI of each projection barcode separately. We adopted a minimal Hamming distance of 2 for the MULTIseq.align function to improve the matching accuracy between detected and designed barcodes. Tag parameters in 'MULTIseq. preProcess' function were adjusted according to our user-defined position of index barcode length

and position. Based on our primer design, the expected format is: cell barcode in Read 1 (bases 1–16), UMI in Read 1 (bases 17–28), and projection barcode in Read 2 (bases 31–45).

## scRNA-seq transcriptional expression analysis

The filtered count matrix was analyzed and processed using Seurat and Scanpy, including data filtering, normalization, highly variable genes selection, scaling, dimension reduction, and clustering (*Stuart et al., 2019*; *Wolf et al., 2018*). First, scRNA-seq data from three samples of unsorted cells and one sample of sorted EGFP-positive cells were created as Seurat object separately; genes with less than three counts were removed and cells with fewer than 200 genes detected were removed. Second, four Seurat objects were merged using the 'merge' function in Seurat. Downstream analysis of merged Seurat objects were as follows: (1) Data filtering: cells with a mitochondrial gene ratio of greater than 20% were excluded. We kept cells for which we detected between 500 and 8000 genes (cells with more than 8000 genes detected were considered potential doublets), and between 1000 and 60,000 counts (cells with more than 60,000 counts detected were considered potential doublets). (2) Data normalization: for each cell, counts were log normalized with the 'NormalizeData' function in Seurat; 'scale.factor' was set to 50,000. (3) Highly variable gene selection: 2000 highly variable genes were calculated using the 'FindVariableFeatures' function in Seurat. (4) Data scaling: the Seurat object was performed using the 'ScaleData' function with default parameters. The number of counts, number of genes, mitochondrial gene ratio, and sorting condition were regressed out in 'ScaleData'. (5) Principal component analysis: highly variable genes were used to calculate principal components in the 'RunPCA' function. A total of 100 principal components (PCs) were obtained and stored in Seurat object for computing neighborhood graphs and uniform manifold approximation and projection (umap) in following section. (6) Leiden clustering: Seurat object was converted into loom file and imported by Scanpy. A neighborhood graph of observations was computed by 'scanpy.pp.neighbors' function in Scanpy. Then, leiden algorithm was used to cluster cells by 'scanpy.tl.leiden' function in Scanpy. (7) Cluster merge and trimming: The top 200 DEGs for each cluster were calculated using the 'scanpy.tl.rank_genes_groups' function in Scanpy using parameters method='wilcoxon' and n_genes = 200. Cluster annotation was performed manually based on previously reported markers of PFC all cell types, layer, neuron subtypes, and mouse brain atlas (*Bhattacherjee et al., 2019*; *Sorensen et al., 2015*). Cell clusters with similar marker genes were merged into one cluster. Complete marker lists for all cell types and all excitatory neuron subtypes calculated using 'FindAllMarkers' function in Seurat were provided (see *Supplementary files 2 and 3*).

Two rounds of clustering were performed. In the first round, we clustered all cells detected by scRNA-seq to generate major cell type classification, that is excitatory neurons, inhibitory neurons, astrocytes, oligodendrocytes, endothelial cells, and microglia. Then we use the annotated 'Excitatory neuron' cluster to further cluster excitatory neuronal subtypes. In the 2nd round clustering, we found several clusters expressed a lower number of counts per cell, a lower number of genes per cell, a higher percentage of mitochondria genes, and ribosome protein genes as DEGs, which indicates cell clusters with low cell quality (*Ilicic et al., 2016*). We also found several other clusters with a small number of cells expressing typical markers of non-neuron cells, such as microglia (*C1qa*, *C1qb*) oligodendrocytes (*Olig1*, *Olig2*) and endothelial cells (*Flt1*, *Cldn5*), which indicated 'contamination' of other cell types mixed in 'Excitatory neuron' in the initial clustering results. We then filtered out those cells from 'Excitatory neuron' cluster and redid clustering to generate excitatory neuronal subtypes (see *Supplementary file 1*).

## Cell type correspondence assessment

To evaluate whether the transcriptional cell types we recovered and annotated correlated with cell types from spatial transcriptomics of PFC or other scRNA-seq datasets of PFC, we used a previously reported comparison analysis method (*Bhattacherjee et al., 2023*). Briefly, we integrated our dataset and previously reported datasetes (*Bhattacherjee et al., 2023*; *Bhattacherjee et al., 2019*; *Lui et al., 2021*; *Yao et al., 2021*) into a harmonized PCA space using the Harmony algorithm (*Korsunsky et al., 2019*). We then constructed a K-nearest neighbor (KNN) graph incorporating all cells from the two datasets. We used the first 30 harmonized principal components as inputs for FindNeighbors function of Seurat to calculate the KNN. For each cluster of public dataset, we found its 30 nearest neighbor cells and determined the percentages of those cells belonging to each scRNA-seq cluster of our

dataset. This created a correspondence matrix showing the transcriptional similarity of each public dataset cluster to each cluster of our dataset.

In this matrix, rows represent our scRNA-seq clusters, columns represent public dataset clusters, and the matrix values reflect the degree of similarity between the clusters. This process was reciprocally conducted for clusters of our dataset, comparing them to public dataset clusters to form a secondary correspondence matrix. The mean of these two matrices provided a quantifiable measure of the similarity between cell clusters identified by our annotation and public dataset annotation.

## Binary projection pattern classification

To determine valid barcoded cells, we first calculated the 95th percentile of the total number of unique molecular identifiers (nUMI) that were mapped with five barcodes, and removed the unusually high numbers of UMIs, which might indicate doublets or PCR-biased amplification. Next, we used two set of cells as negative control, that is, cells supposed not to contain projection barcodes. First set of negative control cells we used is non-neuronal cells classified by coarse clustering based on single-cell transcriptome (*Tervo et al., 2016*). Second set of negative control cells we used is 'EGFP-negative' cells in FAC-sorted dataset. Basically, we calculated the total five projection barcodes counts determined by cellranger of FAC-sorted dataset, then we assigned the cells with zero projection barcodes (nUMI of *EGFP* RNA = 0) counts as 'EGFP-negative' cells. For two set of negative control cells, we searched for the value in the empirical cumulative distribution function (ECDF) that is closest to the 99.9th percentile agains each projection barcode, respectively. We selected the higher UMI threshold from the two given sets of threshold values. A cell is determined to be validly barcoded if the number of the barcode UMIs within the cell is larger than the threshold. For example, the calculated threshold of UMIs for barcode 0 (AI) is 28, which means if a cell contains more than 28 UMIs of barcode 0, then this cell is validly barcoded by AI. UMIs threshold for DMS, 101; for MD, 114; for BLA, 35; for LH, 103. Finally, we dropped UMI counts of determined non-barcoded cells to zero to obtain the index barcode counts matrix used for downstream analysis. Binary projection patterns were calculated by five projection targets set intersections of corresponding barcoded cells. Only the top 10 frequent binary and collateral projection patterns were kept for reliable inference.

## Projection pattern-specific DEGs analysis

DEGs were calculated using the default parameters of the 'FindMarkers' function in Seurat, except the MAST algorithm was used to do DE testing. For the DEG volcano plot, the chosen cut-off for statistical significance was $10^{-10}$ (*Figure 4* and *Figure 4—figure supplement 1*) or $10^{-5}$ (*Figure 5* and *Figure 5—figure supplement 1*) and chosen cut-off for absolute $\log_2$ fold-change was 0.5. Volcano plots were implemented using the EnhancedVolcano R package (v1.4.0).

For the DEG heatmap in *Figure 5A*, the top 10 DEGs ordered by average $\log_2$ fold-change were chosen from each binary cluster. The heatmap was implemented using the 'scanpy.pl.heatmap' function in Scanpy.

## Joint analysis of MERGE-seq and fMOST projection patterns

Single-neuron projectome data for five PFC target regions (AI, dorsal striatum, BLA, MD, LH) were extracted from *Gao et al., 2022*. Projection patterns were quantified by calculating the percentage of each pattern relative to total patterns. Patterns were categorized by number of targets (1, 2, 3, or ≥3 targets). MERGE-seq and fMOST projection pattern percentages were statistically compared within each category using two-sided Wilcoxon tests with Holm correction for multiple comparisons.

## Machine learning implementation on projection and transcription data

Naïve Bayes was applied to perform a machine learning classification task. We first encoded binary projection labels for each projection target (barcoded and non-barcoded) and five set of models (AI, DMS, BLA, LH and MD) were independently trained. We explored a parameter range of number of the top highly variable genes (HVGs) (2, 5, 10, 20, 50, 100, 200, 300, 400, 500, 1000, 2000, 5000) to fit the model. A total of 1000 cells were randomly sampled from 8210 excitatory neurons and top HVGs were selected by default order of results based on 'FindVariableFeatures' function of Seurat per trial. In total, 100 trials were repeated.

To interpret contribution of important genes for each HVGs-based Naïve Bayes model, data matrix for modeling building was constructed as below: for each projection target, 8210 excitatory neurons × (normalized expression of the top 50 HVGs + binary projection labels), or 8210 excitatory neurons× (normalized expression of 50 random genes +binary projection labels). Each data matrix was shuffled first and split by training-testing data in a ratio of 0.7. Machine learning workflow was implanted in pycaret python package (v2.3.4) 'pycaret.classification' module. First, for each model, we used 'setup' function to initialize the training environment and created the transformation pipeline by setting 'target' parameter to column name of input data matrix corresponding to binary projection labels. Then we used 'create_model' function to train and evaluate the performance of a given model by setting 'estimator' parameter to 'nb' and other parameters by default.

To validate barcode/non-barcode label integrity, we performed 100 iterations of random sampling 1000 cells and swapping barcoded with non-barcoded labels. Prediction accuracy, AUC, and F1 scores were compared between original models using the top 50 HVGs with true labels versus models with swapped labels. For each of the 100 trials, 1000 cells were randomly sampled from the 8210 total cells, and barcoded/non-barcoded labels were swapped to the extent possible based on the smaller group. Models were built for each target using original or swapped labels and the top 50 HVGs.

We implemented kernel explainer of SHAP python package (v0.40.1) to summarize the effects of genes. SHAP explainer was created using 'shap.KernelExplainer(model.predict, training data)' function. SHAP values were calculated using 'explainer.shap_values(testing data)' function, and plotted by 'shap.summary_plot()' function to create a SHAP beeswarm plot by displaying top 20 features. Training data and testing data for calculating SHAP values were subsampled with 1500 cells.

## Statistical analysis

No statistical methods were used to predetermine sample size. The experiments were not randomized and investigators were not blinded to allocation during experiments and outcome assessment. Two-sided Wilcoxon test with Holm correction for multiple comparisons was performed in *Figure 3D*, *Figure 6B*, and *Figure 6—figure supplement 1B*. Detailed summary statistics were provided in corresponding Source data files.

## Acknowledgements

We thank Dr. Liye Zhang, Pin Wu and Hengxin Liu for generous advice on the bioinformatic analyses.

## Additional information

### Funding

| Funder | Grant reference number | Author |
| --- | --- | --- |
| National Natural Science Foundation of China | 92368204 | Yuejun Chen |
| STI2030-Major Projects | 2021ZD0200900 | Yuejun Chen |
| Shanghai Science and Technology Development Funds | 23JS1401400 | Yuejun Chen |
| Strategic Priority Research Program of the Chinese Academy of Sciences | XDB32030200 | Yuejun Chen |
| Shanghai Municipal Science and Technology Major Project | 2018SHZDZX05 | Yuejun Chen |
| National Natural Science Foundation of China | 32170806 | Yuejun Chen |
| National Natural Science Foundation of China | 32130035 | Zhen-Ge Luo |

| Funder | Grant reference number | Author |
| --- | --- | --- |
| Central Guidance on Local Science and Technology Development Fund | YDZX20233100001002 | Zhen-Ge Luo |
| National Key Research and Development Program of China | 2021ZD0202500 | Zhen-Ge Luo |
| National Key Research and Development Program of China | 2019YFA0709504 | Chengyu T Li |
| the Innovations of Science and Technology 2030 from the Ministry of Science and Technology of China | 2021ZD0203601 | Chengyu T Li |
| National Natural Science Foundation of China | 31827803 | Chengyu T Li |
| the Shanghai Municipal Science and Technology Major Project | 2021SHZDZX | Chengyu T Li |
| National Natural Science Foundation of China | 92168107 | Zhen-Ge Luo |
| National Natural Science Foundation of China | 32161133024 | Chengyu T Li |

The funders had no role in study design, data collection and interpretation, or the decision to submit the work for publication.

## Author contributions

Peibo Xu, Conceptualization, Data curation, Software, Formal analysis, Validation, Investigation, Visualization, Methodology, Writing – original draft, Writing – review and editing; Jian Peng, Conceptualization, Resources, Data curation, Investigation, Methodology; Tingli Yuan, Data curation, Investigation, Visualization; Zhaoqin Chen, Resources, Data curation, Investigation; Hui He, Data curation, Validation, Visualization; Ziyan Wu, Data curation; Ting Li, Resources; Xiaodong Li, Software, Investigation; Luyue Wang, Software, Validation, Investigation; Le Gao, Jun Yan, Investigation; Wu Wei, Software, Investigation, Visualization; Chengyu T Li, Resources, Supervision, Funding acquisition, Project administration, Writing – review and editing; Zhen-Ge Luo, Conceptualization, Resources, Supervision, Funding acquisition, Methodology, Project administration, Writing – review and editing; Yuejun Chen, Conceptualization, Resources, Formal analysis, Supervision, Funding acquisition, Investigation, Methodology, Writing – original draft, Project administration, Writing – review and editing

## Author ORCIDs

Peibo Xu ⬤ http://orcid.org/0000-0001-7129-0445
Tingli Yuan ⬤ http://orcid.org/0000-0001-6676-8790
Xiaodong Li ⬤ http://orcid.org/0000-0002-2459-8778
Zhen-Ge Luo ⬤ https://orcid.org/0000-0001-5037-0542
Yuejun Chen ⬤ https://orcid.org/0000-0002-4625-2604

## Ethics

All animal experiments were conducted according to a protocol approved by the IACUC at the Institute of Neuroscience, CAS Center for Excellence in Brain Science and Intelligence Technology of the Chinese Academy of Sciences (Shanghai, China). (reference number for approval: NA-034-2022).

## Decision letter and Author response

Decision letter https://doi.org/10.7554/eLife.85419.sa1
Author response https://doi.org/10.7554/eLife.85419.sa2

## Additional files

### Supplementary files

• Supplementary file 1. Marker genes of 2nd round clustering of "Excitatory neuron" cluster, related to *Figure 2*. The inserted umap shows the number of UMI counts (nCount_RNA) per cluster, number of genes (nFeature_RNA) per cluster, percentage of mitochondria genes (percent_mt) per cluster.

• Supplementary file 2. Marker genes of 7 scRNA-seq clusters from all excitatory neurons, related to *Figure 2*. Table with marker genes for each cluster calculated using Seurat package using Wilcoxon test.

• Supplementary file 3. Marker genes of 8 clusters from all cells, related to *Figure 1*. Table with marker genes for each cluster calculated using Seurat package using Wilcoxon test.

• Supplementary file 4. Median gene detection metrics for different major cell types, related to *Figure 1*.

• MDAR checklist

### Data availability

Raw gene expression, barcode count matrices and metadata are available from the Gene Expression Omnibus (GSE210174). The computational code used in the study is available at GitHub (https://github.com/MichaelPeibo/MERGE-seq-analysis copy archived at *Peibo, 2024*). The data needed to evaluate the conclusions in the paper can be downloaded at https://figshare.com/projects/High-throughput_mapping_of_single-neuron_projection_and_molecular_features_by_retrograde_barcoded_labeling/150207. All data needed to evaluate the conclusions in the paper are present in the paper and/or the Supplementary Materials and source data files.

The following datasets were generated:

| Author(s) | Year | Dataset title | Dataset URL | Database and Identifier |
| --- | --- | --- | --- | --- |
| Xu P, Peng J, Yuan T, Chen Z, Wu Z, Luo ZG, Chen Y, Li CT | 2024 | High-throughput mapping of single-neuron projection and molecular features by retrograde barcoded labeling | https://www.ncbi.nlm.nih.gov/geo/query/acc.cgi?acc=GSE210174 | NCBI Gene Expression Omnibus, GSE210174 |
| Peibo X | 2022 | figure1&S1 | https://doi.org/10.6084/m9.figshare.21298842.v4 | figshare, 10.6084/m9.figshare.21298842.v4 |
| Peibo X | 2022 | figure5&S5 | https://doi.org/10.6084/m9.figshare.21298896.v2 | figshare, 10.6084/m9.figshare.21298896.v2 |
| Peibo X | 2022 | figure4&S4 | https://doi.org/10.6084/m9.figshare.21298893.v2 | figshare, 10.6084/m9.figshare.21298893.v2 |
| Peibo X | 2022 | figure6&S6 | https://doi.org/10.6084/m9.figshare.21298899.v2 | figshare, 10.6084/m9.figshare.21298899.v2 |
| Peibo X | 2022 | figure2&S2 | https://doi.org/10.6084/m9.figshare.21298839.v2 | figshare, 10.6084/m9.figshare.21298839.v2 |
| Peibo X | 2022 | figure3&S3 | https://doi.org/10.6084/m9.figshare.21298884.v2 | figshare, 10.6084/m9.figshare.21298884.v2 |
| Peibo X | 2022 | Fig4_CDEFGH_confocal | https://doi.org/10.6084/m9.figshare.21258366.v1 | figshare, 10.6084/m9.figshare.21258366.v1 |
| Peibo X | 2022 | Fig4E&H_Fig5F_FigS4D_IHC_quant_data | https://doi.org/10.6084/m9.figshare.21256449.v1 | figshare, 10.6084/m9.figshare.21256449.v1 |

The following previously published datasets were used:

| Author(s) | Year | Dataset title | Dataset URL | Database and Identifier |
|---|---|---|---|---|
| Bhattacherjee A, Djekidel MN, Chen R, Chen W, Tuesta LM, Zhang Y | 2019 | Cell type-specific transcriptional programs in mouse prefrontal cortex during adolescence and addiction | https://www.ncbi.nlm.nih.gov/geo/query/acc.cgi?acc=GSE124952 | NCBI Gene Expression Omnibus, GSE124952 |
| Lui JH, Luo L | 2020 | Single cell RNAseq of Rbp4cre+ neurons from prefrontal cortex | https://www.ncbi.nlm.nih.gov/geo/query/acc.cgi?acc=GSE161936 | NCBI Gene Expression Omnibus, GSE161936 |

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
