## [Editor Report]

This manuscript describes a valuable new circuit mapping and profiling technique called Multiplexed projEction neuRons retrograde barcodE (MERGEseq) that combines transcriptome and projectome data at a single-cell resolution. The authors provide solid evidence that MERGEseq can be used to identify projection targets and cell type/layer/transcriptome differences of projection neurons in the mouse prefrontal cortex, and validation experiments are rigorous. While this report is a proof-of-principle that MERGEseq is useful for circuit mapping and profiling and many potential details will influence conclusions, this technique could easily be adapted to other regions with known projection targets and adds to a growing arsenal of combinatorial circuit mapping and profiling tools.

---

## [Decision Letter]

**Decision letter after peer review:**

Thank you for submitting your article "High-throughput mapping of single-neuron projection and molecular features by retrograde barcoded labelling" for consideration by *eLife*. Your article has been reviewed by 3 peer reviewers, one of whom is a member of our Board of Reviewing Editors, and the evaluation has been overseen by Kate Wassum as the Senior Editor. The reviewers have opted to remain anonymous.

The reviewers have discussed their reviews with one another, and the Reviewing Editor has drafted this to help you prepare a revised submission. Please accompany your revision with a point x point response to each point raised by the reviewers in the public and private reviews, paying special attention to the essential revisions noted below.

Essential revisions:

1) The manuscript builds upon several prior approaches, only some of which are discussed and cited. The scholarship of the manuscript needs to be improved by incorporating a more robust review of other approaches, as suggested by reviewers to place these findings in context. The scholarship of the manuscript should also be improved by including a more transparent discussion of the limitations of MERGE-seq relative to other approaches.

2) Reviewers raised several issues related to the efficiency of retrograde labeling and barcode recovery that have the potential to affect the conclusions reached in the manuscript. These issues should be addressed in the manuscript with new data and/or analysis.

3) A revision should incorporate requested information to provide clarity about the experimental details and analysis methods.

4) Please ensure your manuscript complies with the *eLife* policies for statistical reporting: https://reviewer.elifesciences.org/author-guide/full "Report summary statistics (e.g., t, F values) and degrees of freedom, exact p-values, and 95% confidence intervals wherever possible. These should be reported for all key questions and not only when the p-value is less than 0.05.

5) Please include a key resource table.

*Reviewer #1 (Recommendations for the authors):*

1. The introduction discusses current techniques like BARseq but makes no mention of the current retrograde tracing and sequencing techniques which is what MERGEseq is actually improving upon. While these retro techniques are discussed in detail in the discussion, it seems odd that they are left out of the introduction as MERGEseq is really an extension of these techniques rather than MAPseq/BARseq.

2. Figure 1 – Supplement 1 shows that very different UMI cutoffs were used to call a cell "positive" for each barcode index. This would presumably preclude direct comparison across regions with different barcodes, at least for purposes of determining the density of projections (since a different degree of projections will be excluded for each virus region). This should be mentioned more explicitly in the manuscript.

3. Lines 170-171 note that "stressed" 637 neurons were filtered out from the Slc17a7 population. It is not clear what this means, and Figure 2 – Supplement 1A does not explain this. This should be clarified.

4. In the discussion, authors might comment on the added value of capturing the "full" transcriptome at the cost of spatial resolution.

5. It may be good to mention that retro-AAV2 has not been reported to infect fibers of passage.

6. Please include a color scale for Figure 5a.

7. Please include axis labels for Figure 5c.

8. Figure 5F shows the DMS+LH Pou3f+ cells, what about the DMS only and LH only? Are there any Pou3f+ cells that lack fluorescence?

9. Please enlarge Figure 3e.

10. For scRNA-seq, please provide detail on the depth of sequencing, the number of GEMwells used, and the number of technical or biological replicates.

11. Please include a color scale for Figure 6e-f.

12. Typo, line 536 "despite of chosen"; should be "in spite of chosen" or "despite chosen".

13. Supplemental Figure 2 panel E, missing cell type labels going across.

14. Please cite the original retro seq paper (Tasic, 2018, PMCID: PMC6456269).

*Reviewer #2 (Recommendations for the authors):*

Based on their detection of barcodes in single-cell RNA-sequencing, the authors conclude that "about 74% of barcoded vmPFC neurons projected to one of these five targets (dedicated projection) and 26% of barcoded vmPFC neurons sent collateral projections to multiple brain regions…" (lines 92-94 in Introduction, lines 242-244 in Results). These conclusions are contingent upon 100% of neurons that project to a specific region being labeled by retrograde barcoded viruses and barcodes are detected at 100% efficiency. The authors did not provide an estimation of either efficiency. In the penultimate paragraph of the Discussion, the authors raised this as "A potential technical concern" but conclude that "the overall dedicated and collateral projection pattern…will not be greatly affected by the labeling efficiency or recovery rate."

I disagree with the authors' conclusion. Suppose that retrograde labeling efficiency is 70%, and the barcode recovery rate is also 70% (both very optimistic estimates). Suppose further that all neurons of a particular type send collateral branches to two target regions, X and Y. The experiment will yield the results that 25% of the neurons will be labeled by barcodes injected at X only, 25% by barcodes injected at Y only, 25% labeled by both barcodes, and 25% labeled by neither. The conclusion from the above experiment would be that 2/3 of neurons are "dedicated" to either X or Y, and 1/3 of the neurons send axons to both regions. This simple back-of-the-envelope calculation reveals how much collateralization is underestimated by incomplete retrograde labeling!

Without reading the penultimate Discussion paragraph, readers will be misled twice about the fraction of "dedicated projection." Even after reading it, the readers will still be misled by the authors' conclusions.

If the authors wish to make a quantitative conclusion about the true "dedicated" vs. "collateral" projections, they must determine the efficiency of retrograde barcoding. They can inject AAVretro carrying two different types of barcodes into the same region (via two separate injections, rather than injecting a mixture, which will artificially raise the co-transduction efficiency) and quantify individual cells that are labeled by both types of barcodes. They can then use such efficiency to calibrate their estimation of "dedicated" vs. "collateral" projections. (Note that retrograde labeling efficiency may differ for different sites.) Without such calibration, the authors should caution the readers about the (likely large) underestimation of true collateralization whenever such data are presented and discussed.

Another issue with "dedicated" projections: the authors only examined 5 targets. Each of the "dedicated" projections is true within these 5 targets, these neurons can send collateralized axons to other, unstudied targets.

*Reviewer #3 (Recommendations for the authors):*

1) Please provide a better context for the presented method. Clarify how the transcriptomic cell type definition in this paper corresponds to previous papers and clarify how the presented method differs from previous methods and what the advantages and disadvantages are. Please cite other papers that have employed single-cell Retro-seq: Tasic et al. 2016 (https://doi.org/10.1038/nn.4216), Tasic et al. 2018 (https://doi.org/10.1038/s41586-018-0654-5) , Yao et al. 2021, https://doi.org/10.1016/j.cell.2021.04.021, Zhang et al. 2021 https://doi.org/10.1038/s41586-021-03223-w )

2) Line 171: How were 'stressed' neurons defined? Please explain.

3) Please mention which 10x Genomics chemistry version (e.g., v2, v3, v3.1) you use in the main text and criteria for QC (lowest acceptable UMI or gene detection level, as well as median gene detection for different cell classes: excitatory, inhibitory and non-neuronal).

4) Figure 1E: The figure shows a correlation between two studies at a resolution that is not appropriate – too low to be informative except for QC. Please move this to supplement.

5) Cell type identity definition: I suggest performing data integration (for example using Seurat) with Bhattacherjee et al., 2019, Liu et al. 2021 and with Yao et al. 2021 (see above) to give more updated names to cell types. The paper should start with cell type definition and present consistent nomenclature from the beginning.

6) Caution should be exercised when interpreting under-represented cell types in MERGE-seq. Figure 2 shows that only 8 neurons of the L5-Htr2c subtype were validly barcoded. However, in addition to the possibility that this subtype does not project to the five targets included in this study, the small number of barcoded L5-Htr2c subtype could also be caused by the tropism of AAV2-Retro viruses, or the selective loss of L5-Htr2c neurons in tissue processing due to cell death. Comparison with the in situ hybridization patterns of marker genes in Supplementary Figure 2 and the proportions of neuronal subtypes in Figure 2b suggests bias in sampling of cell subtypes by scRNA-seq. Therefore, independent approaches should be utilized to further investigate the projection targets of L5-Htr2c neurons before reaching a conclusion.

7) We suggest replacing "unbarcoded" with "non-barcoded". "Un-barcoded" sounds like the cells were barcoded and then the barcode was removed. The more appropriate term is "non-barcoded".

8) Figure 2 (and others): Please state how many cells are represented in each panel of the figure.

9) Figure 2 E and F – Not the most straightforward and informative representation: We suggest converting these to bar plots per area and per type. It is good to see that the authors kept the colors introduced in this figure in Figure 3. We suggest wherever the color code can be kept consistent, to do so.

10) Figure 3. MERGE-seq reveals hidden projection diversity within the vmPFC – please remove 'hidden'.

11) Figure 6E/F – How many neurons and which types are shown? Every figure should state which single-cell transcriptomes were included and the labeling should be consistent with the previous figures. For example, please show the PC1/PC2 scatter plot in E and F next to the same cells labeled with their cell type assignments + colors. This allows the reader to connect the information from previous figures to these.

12) Line 490/491 Please include these references when referring to Retro-seq: Tasic et al. 2016 (https://doi.org/10.1038/nn.4216), Tasic et al. 2018 (https://doi.org/10.1038/s41586-018-0654-5), Yao et al. 2021, https://doi.org/10.1016/j.cell.2021.04.021, Zhang et al. 2021 https://doi.org/10.1038/s41586-021-03223-w )

13) Completeness and sensitivity of barcode recovery of the approach: The authors recovered 1791 EGFP-positive cells undergoing fluorescence-activated cell sorting (FACS) from three mice and 19,470 single cells without sorting from the other three mice. Using thresholds calculated based on barcode counts in non-neurons, they found that the percentage of barcoded cells in FACS sorted or unsorted groups are 54% and 12%, respectively. Given that almost all cells sorted by FACS should have been infected with the barcoded GFP AAV viruses, the recovery rate for barcodes is low. Therefore, labeling neurons as barcoded and unbarcoded based on the detection of barcodes will create false negatives, that is, classifying many retrograded labeled neurons as "unbarcoded" (we suggest changing this to "non-barcoded"). It seems that the projectomes of neurons cannot be simply derived from barcode detection through sequencing. Many "unbarcoded" projection neurons were in fact retrogradely labeled by AAV viruses injected into a specific target but were negative for barcodes due to technical limitations.

One immediate issue caused by the false negative rate of barcode detection is whether the machine learning-based modeling is provided with the right training data (Figure 6). For this model to predict projectomes based on transcriptomes, it first requires a good correlation between barcoding and projectome.

More discussion should be given to the recovery rate of barcodes, and its potential impact on data analysis.

14) Additional analysis and control experiments related to barcode detection:

The low barcode recovery rate could be due to the low number of copies for AAV-encoded transcripts in the transcriptome of single cells, or it is specific for the detection of short barcode sequences. With the current scRNAseq data, one additional analysis is to measure the percentage of neurons positive for GFP transcripts from both FAC-sorted and non-sorted samples and compare the GFP+ neuron frequency to barcode+ neuron frequency.

15) To enrich cDNA fragments composed of the barcode index, unique molecular identifiers (UMIs), and the barcode, the authors prepared expressed virus barcode libraries with special primers. The authors also tried to detect barcodes directly in single-cell transcriptional libraries. What was the barcode detection frequency without this additional amplification, that is, in regular single-cell transcriptional libraries? It would be good to comment on how much this approach (we assume) improves barcode detection compared to the regular 10x single-cell libraries without additional barcode amplification.

16) The authors hypothesized that the non-neuronal cells would not be transduced by rAAV2-retro. The barcode counts in these non-neuronal cells were used to generate the thresholds for projection neurons. However, such cells, especially microglia, could be positive for AAV transcripts perhaps by phagocytosing dying infected neurons. A better control cell population could be cells negative for GFP after FACS. If sequencing data are available for such GFP-negative cells, it would be useful to examine the detection of barcodes in these cells and use them as the negative control for thresholding.

17) In the section on Projection barcode FASTQ alignment, the authors stated that deMULTIplex R package (v1.0.2) (https://github.com/chris-mcginnis-ucsf/MULTI-seq) was used to count UMIs associated with barcodes. This method was designed to detect the Sample index and needs to be further adjusted for barcode reading. The current method did not reveal the fact that a single UMI could be associated with multiple barcodes, which could raise the need for thresholding at this stage.

MULTIseq.preProcess was used to identify the barcode sequence based on its position relative to the P7 primer. The result is a readTable with each row showing the Cell ID, UMI, and barcode sequence. The same UMI appeared in multiple rows and could have different barcodes.

MULTIseq.align function was used to match the barcode sequence in each row of readTable to the 5 barcodes, and to find the numbers of UMI associated with each barcode in each sample. This function utilizes a minimal Hamming distance of 1 to call a match between barcodes detected in the sequencing samples and the list of designed barcodes. We would suggest a minimal hamming distance of 2. Many of the sequences with a minimal Hamming distance of 2 have a frameshift of 1 nucleotide as compared to the designed barcodes. If the parameter of MULTIseq.preProcess is adjusted to change the position of the expected barcode, we would expect to find the full-length barcode. A specific example is:

"AAGGCACAGACTTTG" has a Hamming distance of 2 as compared to barcode 2 "GAAGGCACAGACTTT", and should also be considered as a match.

More importantly, MULTIseq.align does not consider the complication that multiple barcodes could be detected for the same UMI, and simply uses the barcode of the first duplicated UMI. Taking the FAC-sorted pfc_4 dataset as an example, the readTable generated using this dataset contains 30272582 rows. The third column represents sequences detected at the specified barcode position.

After aligning to the 5 barcodes with a max hamming distance of two, 26443131 of the sequences detected at the specified barcode position can be matched, leading to 474012 unique combinations of Cell/UMI, each combination with a set of detected barcodes.

Many of the UMIs are associated with multiple barcodes. In the pfc_4 dataset, 68955 of the 474012 unique combinations of Cell/UMI are associated with multiple barcodes (14.5%).

When we used the data above to compare the density distributions of barcode counts per UMI per cell for the pfc_4 dataset and that of non-neuronal cells, we find that low barcode counts may not be specific. The best negative control here would be to use GFP-negative cells after FACS.

Depending on the threshold values to eliminate these false barcode counts, we reached an even smaller number of barcoded neurons at the end of the analysis.

[Editors' note: further revisions were suggested prior to acceptance, as described below.]

Thank you for resubmitting your work entitled "High-throughput mapping of single-neuron projection and molecular features by retrograde barcoded labeling" for further consideration by *eLife*. Your revised article has been evaluated by Kate Wassum (Senior Editor) and a Reviewing Editor.

The manuscript has been improved but there are some remaining issues that need to be addressed, as outlined below:

The reviewers have outlined a couple of additional changes that are needed to fully respond to the prior critiques. These represent small changes in the text and, thus, should not present a significant difficulty. Please address these comments in a revised manuscript.

*Reviewer #2 (Recommendations for the authors):*

The revised manuscript has improved. However, the authors still did not address my concern about determining "dedicated" vs. "collateralized" projections. The authors put those numbers in the Introduction and Results without doing the control experiment I suggested (to determine retrograde labeling efficiency) and without mentioning the caveats. The caveat is only mentioned in Discussion (lines 499-500). Readers who missed this sentence will be misled.

Adding a comparison between MERGE-seq and fMOST tracing (the new Figure 3C) is an improvement. As can be seen from the graph, there is a higher percentage of collateralized axons in fMOST compared to MERGE-seq in multiple categories, confirming that MERGE-seq underestimates the fraction of collateralized axons (though to my relief there is not an order-of-magnitude difference).

The authors should add the number of neurons included in the fMOST dataset in the Figure 3C legend.

*Reviewer #4 (Recommendations for the authors):*

This manuscript introduces MERGE-seq, a multiplexed method for profiling transcriptional features of individual neurons projecting to specific targets. The approach involves multiplexed retrograde tracing by injecting distinctly barcoded rAAV-retro viruses into different target areas, followed by scRNAseq of neurons in the source area on the 10xGenomics platform. The projection targets of barcoded neurons in the source area can be inferred by matching the detected barcodes to the barcode sequences to of rAAV-retro viruses injected into the target areas.

Validation of this approach was conducted by injecting rAAVs carrying five distinct 15-nt barcodes to five known ventromedial prefrontal cortex (vmPFC) targets. This revised version has performed integration analysis with previously existing vmPFC scRNA-seq and MERFISH dataset, and compared vmPFC scRNA clusters and the 7 excitatory neuron subtypes analyzed in this study with those in prior datasets. MERGEseq facilitated the identification of vmPFC cell types projecting to distinct areas, revealing that each of the seven identified excitatory neuron subtypes projects to multiple targets, and the five targets receive projections from multiple transcriptomic types. MERGE-seq derived projection patterns were validated through dual-color retro-AAV tracing and were correlated successfully with fMOST-based single-neuron tracing data. Additionally, marker genes for projection-specific cell subclasses were validated in retrogradely labeled vmPFC using RNA FISH for marker detection.

This revised version has effectively tackled the previously raised concerns. Significant efforts have been dedicated to performing an integrated analysis with existing datasets, enhancing the data analysis methodology, and imposing more stringent criteria for barcode determination. The revised manuscript places greater emphasis on acknowledging and incorporating several prior approaches that influenced the development of the MERGE-seq concept. While the efficiency of retrograde barcoding wasn't experimentally addressed by injecting rAAV-retro viruses with different barcodes into the same region, the limitations and potential concerns of MERGE-seq are now explicitly discussed. Additionally, the revised manuscript provides clarity on essential technical aspects, including QC criteria and parameters for evaluating scRNA data quality. In sum, this manuscript is rigorous and thorough, offering a valuable approach for the multiplexed investigation of neuronal transcriptomics and projection targets.

In addition, I suggest that QC criteria should be explicitly listed in the main text. The number of cells passing each QC step should also be listed either in the main text or in the related figures. My understanding is that there is a general QC step for scRNAseq quality based on gene count, total UMI count, and mitochondrial gene expression and that there is another step to identify low-quality cells and contaminated non-neuron cells. It would be very helpful that such information is readily available in the main text.

---

## [Author Response]

Essential revisions:1) The manuscript builds upon several prior approaches, only some of which are discussed and cited. The scholarship of the manuscript needs to be improved by incorporating a more robust review of other approaches, as suggested by reviewers to place these findings in context. The scholarship of the manuscript should also be improved by including a more transparent discussion of the limitations of MERGE-seq relative to other approaches.

We have substantially revised the Introduction to include explicit comparisons of our method to current approaches in neuronal projectome mapping. Additionally, we have expanded the Discussion to transparently address limitations and potential concerns related to our approach, as well as suggested solutions. Throughout, we have ensured proper scholarly attribution by adding citations acknowledging foundational work and related methods developed by others in this emerging field.

2) Reviewers raised several issues related to the efficiency of retrograde labeling and barcode recovery that have the potential to affect the conclusions reached in the manuscript. These issues should be addressed in the manuscript with new data and/or analysis.

We recognize the inherent challenges associated with imperfect retrograde labeling efficiency in retro-seq-based methods, acknowledging that this efficiency can vary across different anatomical sites, which complicates the quantitative determination of projection patterns. In light of this, we have carefully revised the presentation of our results and discussion in the manuscript to explicitly caution readers about drawing quantitative conclusions.

We suggest that integrating single-molecule imaging techniques, such as MERFISH, with MERGE-seq could potentially provide spatially-resolved and quantitatively-enhanced projection pattern data. Regarding barcode recovery, we have incorporated new computational analyses and established more stringent recovery criteria, supplemented by negative controls, to enhance the accuracy of our barcode detection.

Furthermore, we have openly discussed the difficulties in accurately identifying valid projection barcodes, given the imperfect nature of retrograde labeling efficiency, variable recovery rates, and the distinct projection strengths associated with different target regions. Our Discussion section has been expanded to clearly communicate how these technical factors could impact the interpretation of our data, ensuring a more transparent presentation of our study's limitations and findings.

3) A revision should incorporate requested information to provide clarity about the experimental details and analysis methods.

We have enriched experimental and computational details in Methods section. We also made our computational code fully available in Github.

4) Please ensure your manuscript complies with the eLife policies for statistical reporting: https://reviewer.elifesciences.org/author-guide/full "Report summary statistics (e.g., t, F values) and degrees of freedom, exact p-values, and 95% confidence intervals wherever possible. These should be reported for all key questions and not only when the p-value is less than 0.05.

We have included exact p-values and 95% confidence intervals in source data files for corresponding figures where statistical analysis was performed (Figure 3D, Figure 6B, and Figure 6—figure supplement 1B), which contain the full summary statistics.

5) Please include a key resource table.

We have included key resource table at the start of Methods section.

Reviewer #1 (Recommendations for the authors):1. The introduction discusses current techniques like BARseq but makes no mention of the current retrograde tracing and sequencing techniques which is what MERGEseq is actually improving upon. While these retro techniques are discussed in detail in the discussion, it seems odd that they are left out of the introduction as MERGEseq is really an extension of these techniques rather than MAPseq/BARseq.

We have rewritten the Introduction section, with an explicit description of previous methods. In the revised introduction, we reiterate the technological gap in currently existing methods of mapping neuronal projections, including MAPseq (Kebschull et al., Neuron, 2016), BARseq (Chen et al., Cell, 2019; Sun et al., Nature Neuroscience, 2021), ConnectID (Klingler et al., Nature, 2021), BRICseq (Huang et al., Cell, 2020), VECTORseq (Retro-seq-based) (Cheung et al., Cell Rep., 2022). As recommended, we have expanded the Methods to include an in-depth comparison of our approach to other Retro-seq technologies such as Lui et al., 2021, highlighting both technological differences and distinct biological insights. Throughout the revised manuscript, we have also ensured scholarly attribution by incorporating additional references to foundational work and related methods from across this exciting field.

2. Figure 1 – Supplement 1 shows that very different UMI cutoffs were used to call a cell "positive" for each barcode index. This would presumably preclude direct comparison across regions with different barcodes, at least for purposes of determining the density of projections (since a different degree of projections will be excluded for each virus region). This should be mentioned more explicitly in the manuscript.

Downstream target regions can differentially affect projection barcode recovery due to factors including innervation density, projection range, and connection strength. We apply strict UMI thresholds per target calculated based on FAC-sorted neuron barcode quantiles to control for these effects. As suggested, we now explicitly state the distinct UMI cutoffs used for each validated projection and have expanded the Methods to provide full details on our binary projection classification approach (revised Methods, Binary projection pattern classification section).

In the main text of the revised manuscript (line 157), “*It is worth mentioning that the UMI threshold differs for different targets due to different magnitude of barcode expression of each projection target (Figure 1—figure supplement 1F).*”

3. Lines 170-171 note that "stressed" 637 neurons were filtered out from the Slc17a7 population. It is not clear what this means, and Figure 2 – Supplement 1A does not explain this. This should be clarified.

First, we clustered all cells detected by scRNA-seq to generate major cell type classification, i.e., excitatory neurons, inhibitory neurons, astrocytes, oligodendrocytes, endothelial cells, and microglia. Then we use the annotated “Excitatory neuron” cluster to further cluster excitatory neuronal subtypes. Based on the 2nd round clustering result, we found several clusters expressed a lower number of counts per cell, a lower number of genes per cell, a higher percentage of mitochondria genes, and ribosome protein genes as differentially expressed genes, which indicates cell clusters with low cell quality (Ilicic et al., Genome Biology, 2016). We also found several other clusters with a small number of cells expressing typical markers of non-neuron cells, such as microglia (C1qa, C1qb) oligodendrocytes (Olig1, Olig2) and endothelial cells (Flt1, Cldn5), which indicates “contamination” of other cell types mixed in “Excitatory neuron” in the initial clustering results. Based on these analyses, we filtered out those cells from “Excitatory neuron” cluster and redid clustering to generate excitatory neuronal subtypes.

We provided a supplementable table (supplementary file 1) containing the marker genes of 2nd round clustering before filtering out 637 cells in the revised manuscript. We improved the main text for clarity by changing “stressed neurons” to “low-quality cells and contaminated non-neuron cells” and included this detailed analysis description in the revised manuscript.

4. In the discussion, authors might comment on the added value of capturing the "full" transcriptome at the cost of spatial resolution.

We believe that our current methodology has the advantages below. First, we use a commercialized scRNA-seq technology (10X Genomics), which does not require a specialized technique or equipment setup and is cost-inexpensive. Second, there are many elaborate published spatially-resolved mouse brain atlases generated by Allen Brain Institute, Xiaowei Zhuang’s lab, and many others (Zhang et al., Biorxiv, 2023; Yao et al., Biorxiv, 2023; Allen et al., Cell, 2023). Using differentially expressed genes detected by scRNA-seq 10X Genomics, one can easily annotate the cell population spatially. For example, we performed a comparison analysis of our scRNA-seq data with recently published MERFISH data of PFC (Bhattacherjee et al., Nature Neuroscience, 2023), which confidently provides us with the spatial information of neuronal subtypes (revised Figure 2D). Third, current spatially-resolved single-cell transcriptomics technologies are either imaging-based (MERSCOPE) or next-generation sequencing (NGS)-based (Deng et al., Nature Reviews Bioengineering, 2023). Imaging-based spatial transcriptomics enables single-cell or subcellular resolution, however, is currently cost-expensive, limited by the number of genes detected, and biased to a preselected gene panel. On the other hand, NGS-based spatial transcriptomics enables unbiased gene detection, however, many NGS-based spatial technologies do not support the authentic single-cell resolution (e.g., 50um 10x Genomics Visium or 10-50um for DBiT-seq-based methods) (Deng et al., 2023, Nature Reviews Bioengineering). With all these technological considerations, we believe our method stands out as a robust solution and offers an advantageous balance between resolution and scope.

We have included this part of the discussion in the Discussion part of the revised manuscript (line 473-482).

5. It may be good to mention that retro-AAV2 has not been reported to infect fibers of passage.

We have mentioned this point in the discussion part (line 520). “Further, it has been experimentally verified that AAV2 spread is confined to the vicinity of synaptic terminals and does not affect axon fibers in passages, especially as evidenced by retro-AAV injections in the cervical spinal cord (Wang et al., 2018). ”

6. Please include a color scale for Figure 5a.

Done.

7. Please include axis labels for Figure 5c.

Done.

8. Figure 5F shows the DMS+LH Pou3f+ cells, what about the DMS only and LH only? Are there any Pou3f+ cells that lack fluorescence?

We re-do the statistical analysis to include the percentage of Pou3f1+EGFP−tdT−, Pou3f1+EGFP−tdT+, Pou3f1+EGFP+tdT−, and Pou3f1+EGFP+tdT+ cells in Pou3f1+ cells. EGFP+ cells represent neurons projecting to DMS and tdT+ cells represent neurons projecting to LH. Among Pou3f1+ cells, we found that there are about 31.6% ± 13.1% Pou3f1+EGFP−tdT+ cells, 8.89% ± 2.38% Pou3f1+EGFP+tdT− cells, 55.7% ± 10.4% Pou3f1+EGFP+tdT+ cells, and 3.79% ± 2.91% Pou3f1+EGFP-tdT− cells. This indicates Pou3f1-expressing vmPFC neurons preferentially project to LH and DMS+LH than DMS. We have included this analysis in the revised manuscript (revised Figure 5D-G).

9. Please enlarge Figure 3e.

Done.

10. For scRNA-seq, please provide detail on the depth of sequencing, the number of GEMwells used, and the number of technical or biological replicates.

For unsorted samples, we used 3 mice with three GEM wells in one Chromium Single Cell 3' Chip (v3). Among unsorted samples, sample mouse 1 recovered 8040 cells, 447,984,945 read pairs were aligned, mean reads per cell is 55,719, median genes per cell is 2382; sample mouse 2 recovered 7443 cells, 399,187,134 read pairs were aligned, mean reads per cell is 53,632, median genes per cell is 2379; sample mouse 3 recovered 7243 cells, 410,627,696 read pairs were aligned, mean reads per cell is 56,693, median genes per cell is 2385. For FAC-sorted samples, we used 3 mice with one GEM well in one Chromium Single Cell 3' Chip (v3). FAC-sorted sample recovered 2075 cells, 410,434,792 read pairs were aligned, mean reads per cell is 197,799, median genes per cell is 6533. We have included these details in Methods section in the revised manuscript.

11. Please include a color scale for Figure 6e-f.

We have reorganized the Figure 6. We have added color scale to Figure 6D (previous Figure 6E) and 6F (previous Figure 6F).

12. Typo, line 536 "despite of chosen"; should be "in spite of chosen" or "despite chosen".

We have changed this typo in the revised manuscript.

13. Supplemental Figure 2 panel E, missing cell type labels going across.

Done.

14. Please cite the original retro seq paper (Tasic, 2018, PMCID: PMC6456269).

We have added this citation to the revised manuscript.

Reviewer #2 (Recommendations for the authors):Based on their detection of barcodes in single-cell RNA-sequencing, the authors conclude that "about 74% of barcoded vmPFC neurons projected to one of these five targets (dedicated projection) and 26% of barcoded vmPFC neurons sent collateral projections to multiple brain regions…" (lines 92-94 in Introduction, lines 242-244 in Results). These conclusions are contingent upon 100% of neurons that project to a specific region being labeled by retrograde barcoded viruses and barcodes are detected at 100% efficiency. The authors did not provide an estimation of either efficiency. In the penultimate paragraph of the Discussion, the authors raised this as "A potential technical concern" but conclude that "the overall dedicated and collateral projection pattern…will not be greatly affected by the labeling efficiency or recovery rate."I disagree with the authors' conclusion. Suppose that retrograde labeling efficiency is 70%, and the barcode recovery rate is also 70% (both very optimistic estimates). Suppose further that all neurons of a particular type send collateral branches to two target regions, X and Y. The experiment will yield the results that 25% of the neurons will be labeled by barcodes injected at X only, 25% by barcodes injected at Y only, 25% labeled by both barcodes, and 25% labeled by neither. The conclusion from the above experiment would be that 2/3 of neurons are "dedicated" to either X or Y, and 1/3 of the neurons send axons to both regions. This simple back-of-the-envelope calculation reveals how much collateralization is underestimated by incomplete retrograde labeling!Without reading the penultimate Discussion paragraph, readers will be misled twice about the fraction of "dedicated projection." Even after reading it, the readers will still be misled by the authors' conclusions.If the authors wish to make a quantitative conclusion about the true "dedicated" vs. "collateral" projections, they must determine the efficiency of retrograde barcoding. They can inject AAVretro carrying two different types of barcodes into the same region (via two separate injections, rather than injecting a mixture, which will artificially raise the co-transduction efficiency) and quantify individual cells that are labeled by both types of barcodes. They can then use such efficiency to calibrate their estimation of "dedicated" vs. "collateral" projections. (Note that retrograde labeling efficiency may differ for different sites.) Without such calibration, the authors should caution the readers about the (likely large) underestimation of true collateralization whenever such data are presented and discussed.

We thank reviewer for the constructive suggestion concerning the efficiency of retrograde barcoding in our study. We understand the critical importance of accurately determining the efficiency of retrograde barcoding and acknowledge the challenges presented by varying labeling efficiencies across different sites. Our analysis has focused on cells identified through a stringent barcoding threshold, ensuring reliable data, yet we acknowledge that this approach may not represent the full diversity of neuronal projection patterns.

In response, we have amended the manuscript to more clearly highlight the potential underestimation of collateral projections due to incomplete labeling (line 500-501). This cautionary note emphasizes that our quantitative conclusions might not fully capture the true breadth of neuronal projections. We aim to ensure that readers are thoroughly informed about this limitation and the potential implications for interpreting our data. These revisions have been incorporated into both the Results and Discussion sections to avert any misinterpretation of our study's outcomes. We also propose to combine single-molecule imaging techniques (e.g., MERFISH) with MERGE-seq in the future to generate spatially-resolved and quantitively-enhanced projection patterns (line 517-518). Finally, we provided additional joint analysis with fMOST-based single-neuron projectome data (Gao et al., 2022, Nature Neuroscience) to further validate the projection patterns (> = 3 targets) that cannot be easily validated with dual-color retro-AAV tracing.

Another issue with "dedicated" projections: the authors only examined 5 targets. Each of the "dedicated" projections is true within these 5 targets, these neurons can send collateralized axons to other, unstudied targets.

We have added the claim in the revised manuscript (line 101, 294) claiming the “dedicated” projections are only defined within the five targets examined in this study, and we cannot rule out the possibility of collateral projections of “dedicated” projections defined in this study due to the limited targets we investigated.

Reviewer #3 (Recommendations for the authors):1) Please provide a better context for the presented method. Clarify how the transcriptomic cell type definition in this paper corresponds to previous papers and clarify how the presented method differs from previous methods and what the advantages and disadvantages are. Please cite other papers that have employed single-cell Retro-seq: Tasic et al. 2016 (https://doi.org/10.1038/nn.4216), Tasic et al. 2018 (https://doi.org/10.1038/s41586-018-0654-5) , Yao et al. 2021, https://doi.org/10.1016/j.cell.2021.04.021, Zhang et al. 2021 https://doi.org/10.1038/s41586-021-03223-w )

We have rewritten the discussion part, with a thorough comparison between MERGE-seq and previous methods. We have cited these papers in the revised manuscript.

2) Line 171: How were 'stressed' neurons defined? Please explain.

First, we clustered all cells detected by scRNA-seq to generate major cell type classification, i.e., excitatory neurons, inhibitory neurons, astrocytes, oligodendrocytes, endothelial cells, and microglia. Then we use the annotated “Excitatory neuron” cluster to further cluster excitatory neuronal subtypes. Based on the 2nd round clustering result, we found several clusters expressed a lower number of counts per cell, a lower number of genes per cell, a higher percentage of mitochondria genes and ribosome protein genes as differentially expressed genes, which indicates cell clusters with low cell quality (Ilicic et al., Genome Biology, 2016). We also found several other clusters with a small number of cells expressing typical markers of non-neuron cells, such as microglia (C1qa, C1qb) oligodendrocytes (Olig1, Olig2) and endothelial cells (Flt1, Cldn5), which indicates “contamination” of other cell types mixed in “Excitatory neuron” in the initial clustering results. Based on these analyses, we filtered out those cells from the “Excitatory neuron” cluster and redid clustering to generate excitatory neuronal subtypes. We provided a supplementable tale (revised supplementary file 1) containing the marker genes of 2nd round clustering before filtering out 637 cells in the revised manuscript. We improved the main text for clarity by changing “stressed neurons” to “low-quality cells and contaminated non-neuron cells” and included this detailed analysis description in the revised manuscript.

3) Please mention which 10x Genomics chemistry version (e.g., v2, v3, v3.1) you use in the main text and criteria for QC (lowest acceptable UMI or gene detection level, as well as median gene detection for different cell classes: excitatory, inhibitory and non-neuronal).

We used Chromium Single Cell 3' Reagent Kit (v3) in this study (Cat#PN1000075). We retained cells with a gene count between 500 and 8000, a total UMI count between 1,000 and 60,000, and with less than 20% mitochondrial gene expression, ensuring single-cell data quality and the exclusion of potential outliers. We added the 10x Genomics chemistry version information in the revised Methods part and included median gene detection metrics for different major cell types in the revised supplementary file 4.

4) Figure 1E: The figure shows a correlation between two studies at a resolution that is not appropriate – too low to be informative except for QC. Please move this to supplement.

We have implemented the analytical comparison approach from Bhattacherjee et al., 2023 to evaluate the replication of major cell types between our scRNA-seq data and publicly available spatial references. We replaced the prior analysis in Figure 1E with this improved comparative analysis, now shown in revised Figure 1E for consistency. We consider that it is equally important to demonstrate correspondence across both major classes and excitatory neuronal subtypes.

5) Cell type identity definition: I suggest performing data integration (for example using Seurat) with Bhattacherjee et al., 2019, Liu et al. 2021 and with Yao et al. 2021 (see above) to give more updated names to cell types. The paper should start with cell type definition and present consistent nomenclature from the beginning.

In the revised manuscript, we first performed a comparison analysis of our scRNA-seq data with recently published MERFISH data of PFC (Bhattacherjee et al., 2023, Nature Neuroscience), which confidently provides us with the spatial information of neuronal subtypes (revised Figure 2D). Then we also compare our dataset with the exciatory neuronal subtypes datasets of Bhattacherjee et al., 2019, Lui et al., 2021 and Yao at al., 2021. One thing that is worth mentioning is Lui et al., 2021 sequenced Rbp4cre+ neurons, which are most Layer 5 excitatory projection neurons. Based on the correspondence matrix generated based on MERFISH data and our scRNA-seq data, we have mentioned this spatial nomenclature in the corresponding main text. However, we kept our original annotation to keep the authenticity of the data since we do not have the spatial transcriptomic data. We have included these changes in the revised Figure 2D and revised Figure 2—figure supplement 1D.

6) Caution should be exercised when interpreting under-represented cell types in MERGE-seq. Figure 2 shows that only 8 neurons of the L5-Htr2c subtype were validly barcoded. However, in addition to the possibility that this subtype does not project to the five targets included in this study, the small number of barcoded L5-Htr2c subtype could also be caused by the tropism of AAV2-Retro viruses, or the selective loss of L5-Htr2c neurons in tissue processing due to cell death. Comparison with the in situ hybridization patterns of marker genes in Supplementary Figure 2 and the proportions of neuronal subtypes in Figure 2b suggests bias in sampling of cell subtypes by scRNA-seq. Therefore, independent approaches should be utilized to further investigate the projection targets of L5-Htr2c neurons before reaching a conclusion.

We agree that a small number of barcoded L5-Htr2c neurons could be due to cell loss during single-cell dissociation or tropism selection of AAV2-retro. We have explicitly mentioned this in the revised manuscript (line 202-204) and discussed this in the Discussion part (line 488-495). As suggested by comment 17, we refined our threshold selection criteria and set the Hamming distance threshold to 2. Under the new threshold, we found 9 L5-Htr2c neurons were determined as barcoded.

line 202-204: “Only 9 neurons of the L5-Htr2c subtype were recovered with valid barcodes, which may be attributable to technical factors including cell loss during dissociation or AAV2-retro tropism. Alternatively, this subtype may intrinsically lack projections to the selected target regions examined in this study.”

line 488-495: “For example, only 9 neurons of the L5-Htr2c subtype were recovered with valid barcodes, which may be attributable to technical factors including cell loss during dissociation or AAV2-retro tropism. Alternatively, this subtype may intrinsically lack projections to the selected target regions examined in this study. Furthermore, single-cell dissociation for scRNA-seq can result in cell loss, thereby reducing the recovery rate of barcoded neurons. All these factors could influence the extent to which the complete range of neuronal projections is captured. Consequently, the quantitative conclusions drawn here might not fully represent the true extent of neuronal projections.”

7) We suggest replacing "unbarcoded" with "non-barcoded". "Un-barcoded" sounds like the cells were barcoded and then the barcode was removed. The more appropriate term is "non-barcoded".

We have edited “unbarcoded” into “non-barcoded” across the manuscript.

8) Figure 2 (and others): Please state how many cells are represented in each panel of the figure.

We have clarified the cell number in the figure legend of Figure 1-6 in the revised manuscript.

9) Figure 2 E and F – Not the most straightforward and informative representation: We suggest converting these to bar plots per area and per type. It is good to see that the authors kept the colors introduced in this figure in Figure 3. We suggest wherever the color code can be kept consistent, to do so.

We have transformed the original result into bar plots (revised Figure 2F and 2G).

10) Figure 3. MERGE-seq reveals hidden projection diversity within the vmPFC – please remove 'hidden'.

We have removed “hidden” in the revised manuscript.

11) Figure 6E/F – How many neurons and which types are shown? Every figure should state which single-cell transcriptomes were included and the labeling should be consistent with the previous figures. For example, please show the PC1/PC2 scatter plot in E and F next to the same cells labeled with their cell type assignments + colors. This allows the reader to connect the information from previous figures to these.

As suggested by comment 8, we have added the number of neurons in the respective figures or figures legend. In the Figure 6E and 6F (now as revised Figure 6D and 6F), 1,853 barcoded neurons (top 10 frequent projection patterns) were represented. We want to suggest that at mRNA level, projection target is correlated with certain genes. Thus we did not show the cell type assignments. However, the audience can find corresponding cell type assignments and labels in revised Figure 2E, where we already indicated the distribution of neuronal subtypes for barcoded neurons associated with each projection target.

12) Line 490/491 Please include these references when referring to Retro-seq: Tasic et al. 2016 (https://doi.org/10.1038/nn.4216), Tasic et al. 2018 (https://doi.org/10.1038/s41586-018-0654-5), Yao et al. 2021, https://doi.org/10.1016/j.cell.2021.04.021, Zhang et al. 2021 https://doi.org/10.1038/s41586-021-03223-w )

We have cited these papers in the revised manuscript.

13) Completeness and sensitivity of barcode recovery of the approach: The authors recovered 1791 EGFP-positive cells undergoing fluorescence-activated cell sorting (FACS) from three mice and 19,470 single cells without sorting from the other three mice. Using thresholds calculated based on barcode counts in non-neurons, they found that the percentage of barcoded cells in FACS sorted or unsorted groups are 54% and 12%, respectively. Given that almost all cells sorted by FACS should have been infected with the barcoded GFP AAV viruses, the recovery rate for barcodes is low. Therefore, labeling neurons as barcoded and unbarcoded based on the detection of barcodes will create false negatives, that is, classifying many retrograded labeled neurons as "unbarcoded" (we suggest changing this to "non-barcoded"). It seems that the projectomes of neurons cannot be simply derived from barcode detection through sequencing. Many "unbarcoded" projection neurons were in fact retrogradely labeled by AAV viruses injected into a specific target but were negative for barcodes due to technical limitations.One immediate issue caused by the false negative rate of barcode detection is whether the machine learning-based modeling is provided with the right training data (Figure 6). For this model to predict projectomes based on transcriptomes, it first requires a good correlation between barcoding and projectome.More discussion should be given to the recovery rate of barcodes, and its potential impact on data analysis.

We appreciate the comment of the reviewer. To achieve a stringent categorization of barcoded cells, we set several thresholds to filter cells in order. First, we calculated the 95th percentile of the total number of unique molecular identifiers (nUMI) that are mapped with five barcodes, and removed the unusually high numbers of UMIs, which might indicate doublets or PCR-biased amplification. Next, we used two set of cells as negative control, that is, cells supposed not to contain projection barcodes. First set of negative control cells we used is non-neuronal cells classified by coarse clustering based on single-cell transcriptome. Second set of negative control cells we used is “EGFP-negative” cells in FAC-sorted dataset. Basically, we calculated the total five projection barcodes counts determined by cellranger of FAC-sorted dataset, then we assigned the cells with zero projection barcodes counts as “EGFP-negative” cells. For two set of negative control cells, we searched for the value in the empirical cumulative distribution function (ECDF) that is closest to the 99.9th percentile agains each projection barcode, respectively. We selected the higher UMI threshold from the two given sets of threshold values. Next, a cell is determined to be validly barcoded if the number of the barcode UMIs within the cell is larger than the threshold. For example, the calculated threshold of UMIs for barcode 0 (AI) is 28, which means if a cell contains more than 28 UMIs of barcode 0, then this cell is validly barcoded by AI. UMIs threshold for DMS, 101; for MD, 114; for BLA, 35; for LH, 103. It is worth mentioning that the UMI threshold differs for different targets due to different magnitude of barcode expression of each projection target (Figure 1—figure supplement 1I).

To validate barcode/non-barcode label integrity for machine learning, we performed 100 iterations randomly sampling 1000 cells and swapping labels between barcoded/non-barcoded groups. Prediction accuracy, AUC, and F1 scores of original models using the top 50 HVGs with true labels were compared to models with swapped labels. In each of the 100 trials, 1000 of the 8210 total cells were sampled and barcoded/non-barcoded labels swapped as extensively as group size allowed. We found model performance decreased after label swapping compared to original training labels (revised Figure 6—figure supplement 1B). This analysis suggests that while our stringent UMI threshold might result in some false positives, the barcoded versus non-barcoded labeling approach we adopted still offers a reliable foundation for machine learning training data. Additionally, we have clarified that we limited our analysis and conclusions to data derived from barcoded samples. Line 167-170: *“ The lower fraction of barcoded versus EGFP+ cells suggests our conservative threshold increases false negatives, classifying some low UMI cells as non-barcoded. Therefore, we focused analyses on reliably barcoded cells, though conclusions may not capture the full heterogeneous projection repertoire.”*

We have included the detailed description of threshold determination in the revised manuscript. We also thoroughly discussed the current limitations, computational challenges, potential impact on data analysis, and potential solutions in the revised Discussion (line 484-517).

14) Additional analysis and control experiments related to barcode detection:The low barcode recovery rate could be due to the low number of copies for AAV-encoded transcripts in the transcriptome of single cells, or it is specific for the detection of short barcode sequences. With the current scRNAseq data, one additional analysis is to measure the percentage of neurons positive for GFP transcripts from both FAC-sorted and non-sorted samples and compare the GFP+ neuron frequency to barcode+ neuron frequency.

We appreciate the reviewer’s suggestion. We set the threshold for scRNA-seq data: if GFP UMI counts (sum of all five barcodes UMI counts) in a certain cell are larger than 0, we will determine this cell as a GFP+ cell, otherwise, it is a GFP- cell. We calculated the GFP+ ratio in FAC-sorted and non-sorted groups (Non-sorted group 1-3, group1: 27%, group2: 25%, group3: 27%; FAC-sorted group 4: 81%). In parallel, we also calculated the barcoded ratio in FAC-sorted and non-sorted groups (Non-sorted group 1-3, group1: 17%, group2: 20%, group3: 19%; FAC-sorted group 4: 49%). We can see that barcoded cells ratio is relatively lower than EGFP+ cells. The lower fraction of barcoded versus EGFP+ cells suggests our threshold increases false negatives, classifying some low UMI cells as non-barcoded. Therefore, we focused analyses on reliably barcoded cells, though conclusions may not capture the full heterogeneous projection repertoire.

We added the GFP+ ratio and barcoded ratio across FAC-sorted and non-sorted groups into the revised Figure S1G and S1H, we also discussed this potential limitation in the Discussion section (line 496-517).

15) To enrich cDNA fragments composed of the barcode index, unique molecular identifiers (UMIs), and the barcode, the authors prepared expressed virus barcode libraries with special primers. The authors also tried to detect barcodes directly in single-cell transcriptional libraries. What was the barcode detection frequency without this additional amplification, that is, in regular single-cell transcriptional libraries? It would be good to comment on how much this approach (we assume) improves barcode detection compared to the regular 10x single-cell libraries without additional barcode amplification.

The projectome and scRNA-seq libraries have different structures due to their distinct amplification templates. The projectome library uses full-length post-amplified cDNA, while the scRNA-seq library uses fragmented cDNA. A key advantage of PCR-enriching the EGFP-barcode region from full-length cDNA is that it generates an organized projectome library structure. Based on our primer design, the expected format is: cell barcode in Read 1 (bases 1-16), UMI in Read 1 (bases 17-28), and projection barcode in Read 2 (bases 31-45). In contrast, in the scRNA-seq libraries, the randomly fragmented cDNA produces variable read lengths covering the 15 bp projection barcode, decreasing detection sensitivity and accuracy. Therefore, a direct comparison of projectome and scRNA-seq libraries for barcode detection may not be feasible.

We have calculated the reads that are mapped to the five barcodes, either based on scRNA-seq libraries fastq file using cellranger or based on projectome libraries using our methods adapted from MULTI-seq (McGinnis et al., 2019, Nature Methods). We found that scRNA-seq libraries contain lots of zero counts for projection barcode (Note that we used UMI+1 for plot) (revised FigureS1D).

16) The authors hypothesized that the non-neuronal cells would not be transduced by rAAV2-retro. The barcode counts in these non-neuronal cells were used to generate the thresholds for projection neurons. However, such cells, especially microglia, could be positive for AAV transcripts perhaps by phagocytosing dying infected neurons. A better control cell population could be cells negative for GFP after FACS. If sequencing data are available for such GFP-negative cells, it would be useful to examine the detection of barcodes in these cells and use them as the negative control for thresholding.

We appreciate the rigorous suggestions of the reviewer. We agree with the reviewer that some of the non-neuronal cells could be “barcoded”. In the revised Figure 1F, we have shown that, non-neurons such as endothelial cells, oligodendrocyte progenitor cells and oligodendrocytes contain several cells that have been determined as barcoded. In the revised projection barcode threshold analysis, we also used “EGFP-negative” cells as determined by scRNA-seq data (see revised Methods part, or response to comment 13 for detail). Unfortunately, we did not collect and sequence the GFP-negative cells after FACS. We also agreed that EGFP-negative cells by FACS are an alternative good negative control cells. We have included this point in the Discussion part of the revised manuscript (line 496-517).

17) In the section on Projection barcode FASTQ alignment, the authors stated that deMULTIplex R package (v1.0.2) (https://github.com/chris-mcginnis-ucsf/MULTI-seq) was used to count UMIs associated with barcodes. This method was designed to detect the Sample index and needs to be further adjusted for barcode reading. The current method did not reveal the fact that a single UMI could be associated with multiple barcodes, which could raise the need for thresholding at this stage.MULTIseq.preProcess was used to identify the barcode sequence based on its position relative to the P7 primer. The result is a readTable with each row showing the Cell ID, UMI, and barcode sequence. The same UMI appeared in multiple rows and could have different barcodes.MULTIseq.align function was used to match the barcode sequence in each row of readTable to the 5 barcodes, and to find the numbers of UMI associated with each barcode in each sample. This function utilizes a minimal Hamming distance of 1 to call a match between barcodes detected in the sequencing samples and the list of designed barcodes. We would suggest a minimal hamming distance of 2. Many of the sequences with a minimal Hamming distance of 2 have a frameshift of 1 nucleotide as compared to the designed barcodes. If the parameter of MULTIseq.preProcess is adjusted to change the position of the expected barcode, we would expect to find the full-length barcode. A specific example is:"AAGGCACAGACTTTG" has a Hamming distance of 2 as compared to barcode 2 "GAAGGCACAGACTTT", and should also be considered as a match.More importantly, MULTIseq.align does not consider the complication that multiple barcodes could be detected for the same UMI, and simply uses the barcode of the first duplicated UMI. Taking the FAC-sorted pfc_4 dataset as an example, the readTable generated using this dataset contains 30272582 rows. The third column represents sequences detected at the specified barcode position.After aligning to the 5 barcodes with a max hamming distance of two, 26443131 of the sequences detected at the specified barcode position can be matched, leading to 474012 unique combinations of Cell/UMI, each combination with a set of detected barcodes.Many of the UMIs are associated with multiple barcodes. In the pfc_4 dataset, 68955 of the 474012 unique combinations of Cell/UMI are associated with multiple barcodes (14.5%).When we used the data above to compare the density distributions of barcode counts per UMI per cell for the pfc_4 dataset and that of non-neuronal cells, we find that low barcode counts may not be specific. The best negative control here would be to use GFP-negative cells after FACS.Depending on the threshold values to eliminate these false barcode counts, we reached an even smaller number of barcoded neurons at the end of the analysis.

We thank the reviewer for the careful examination, insightful comments and suggestions regarding the alignment of projection barcode FASTQ in our methods section. We have carefully reviewed and subsequently modified our methodology to address the concerns raised.

First, we have revised the MULTIseq.align function to address the issue of multiple barcodes being detected for the same UMI. We now include a more robust analysis of duplicated UMIs to ensure that all potential barcode matches are considered, rather than defaulting to the first duplicate. Second, we have also adopted a minimal Hamming distance of 2 for the MULTIseq.align function to improve the matching accuracy between detected and designed barcodes. We agree with reviewer that GFP-negative cells after FACS could be an alternatively good control (see response to comment 16). We have now updated our analysis using our improved projection barcode alignment (see Code availability in the revised manuscript) and stringent UMI threshold (see revised Methods part or response to comment 13) to determine barcoded/non-barcoded neurons.

[Editors’ note: what follows is the authors’ response to the second round of review.]

The manuscript has been improved but there are some remaining issues that need to be addressed, as outlined below:The reviewers have outlined a couple of additional changes that are needed to fully respond to the prior critiques. These represent small changes in the text and, thus, should not present a significant difficulty. Please address these comments in a revised manuscript.Reviewer #2 (Recommendations for the authors):The revised manuscript has improved. However, the authors still did not address my concern about determining "dedicated" vs. "collateralized" projections. The authors put those numbers in the Introduction and Results without doing the control experiment I suggested (to determine retrograde labeling efficiency) and without mentioning the caveats. The caveat is only mentioned in Discussion (lines 499-500). Readers who missed this sentence will be misled.

We appreciate your critique on the discussion of the caveats of MERGE-seq. We have now explicitly mentioned the caveats in the Introduction and Results section.

At the end of Introduction, line 104-105:

We revised “Approximately 65% of barcoded vmPFC neurons exhibited dedicated projection patterns” to “Approximately 65% of barcoded vmPFC neurons exhibited dedicated projection patterns based on MERGE-seq data”, clarifying that the conclusion is drawn within the context of MERGE-seq data.

In the Results section, line 258-262: We added “It is worth mentioning that the definition of “dedicated” and “collateral” projections relies solely on the analysis of MERGE-seq data. The quantitative resolution of dedicated and collateral projections of vmPFC neurons will depend on the comprehensiveness of retrograde labeling from all postsynaptic targets and labeling efficiency. ”

Adding a comparison between MERGE-seq and fMOST tracing (the new Figure 3C) is an improvement. As can be seen from the graph, there is a higher percentage of collateralized axons in fMOST compared to MERGE-seq in multiple categories, confirming that MERGE-seq underestimates the fraction of collateralized axons (though to my relief there is not an order-of-magnitude difference).

We appreciate your recognition of our analysis of fMOST data.

The authors should add the number of neurons included in the fMOST dataset in the Figure 3C legend.

We have added the number of neurons in the Figure 3C.

Reviewer #4 (Recommendations for the authors):This manuscript introduces MERGE-seq, a multiplexed method for profiling transcriptional features of individual neurons projecting to specific targets. The approach involves multiplexed retrograde tracing by injecting distinctly barcoded rAAV-retro viruses into different target areas, followed by scRNAseq of neurons in the source area on the 10xGenomics platform. The projection targets of barcoded neurons in the source area can be inferred by matching the detected barcodes to the barcode sequences to of rAAV-retro viruses injected into the target areas.Validation of this approach was conducted by injecting rAAVs carrying five distinct 15-nt barcodes to five known ventromedial prefrontal cortex (vmPFC) targets. This revised version has performed integration analysis with previously existing vmPFC scRNA-seq and MERFISH dataset, and compared vmPFC scRNA clusters and the 7 excitatory neuron subtypes analyzed in this study with those in prior datasets. MERGEseq facilitated the identification of vmPFC cell types projecting to distinct areas, revealing that each of the seven identified excitatory neuron subtypes projects to multiple targets, and the five targets receive projections from multiple transcriptomic types. MERGE-seq derived projection patterns were validated through dual-color retro-AAV tracing and were correlated successfully with fMOST-based single-neuron tracing data. Additionally, marker genes for projection-specific cell subclasses were validated in retrogradely labeled vmPFC using RNA FISH for marker detection.This revised version has effectively tackled the previously raised concerns. Significant efforts have been dedicated to performing an integrated analysis with existing datasets, enhancing the data analysis methodology, and imposing more stringent criteria for barcode determination. The revised manuscript places greater emphasis on acknowledging and incorporating several prior approaches that influenced the development of the MERGE-seq concept. While the efficiency of retrograde barcoding wasn't experimentally addressed by injecting rAAV-retro viruses with different barcodes into the same region, the limitations and potential concerns of MERGE-seq are now explicitly discussed. Additionally, the revised manuscript provides clarity on essential technical aspects, including QC criteria and parameters for evaluating scRNA data quality. In sum, this manuscript is rigorous and thorough, offering a valuable approach for the multiplexed investigation of neuronal transcriptomics and projection targets.

We appreciate your positive assessment of MERGE-seq and recognition of our revised manuscript.

In addition, I suggest that QC criteria should be explicitly listed in the main text. The number of cells passing each QC step should also be listed either in the main text or in the related figures. My understanding is that there is a general QC step for scRNAseq quality based on gene count, total UMI count, and mitochondrial gene expression and that there is another step to identify low-quality cells and contaminated non-neuron cells. It would be very helpful that such information is readily available in the main text.

We appreciate your suggestions on specifically indicating QC criteria in the main text. In the revised manuscript, we have made the following changes:

For the general QC, in the Line 137-142:

We added “We detected 24,788 cells in the raw data matrix. Following initial quality control, which ensured the number of detected RNA in each cell ranged between 500 and 8000, RNA UMI counts in each cell were within 1000 to 60000, and the percentage of mitochondrial genes remained below 20%, we recovered 1791 cells undergoing fluorescence-activated cell sorting (FACS) from three mice and 19,470 single cells without sorting from the other three mice, a total of 21,261 cells.”

For the QC of low-quality cells and contaminated non-neuron cells in the excitatory neurons, in the Line 187-195:

We added “We first re-clustered excitatory projection neurons expressing Slc17a7 (also known as vesicular glutamate transporter, Vglut1). Clusters with low gene/UMI counts and high mitochondrial gene expression were filtered out as low-quality (Ilicic et al., 2016). Some clusters exhibited non-neuronal cell markers like microglial genes (C1qa, C1qb), oligodendrocyte genes (Olig1, Olig2), and endothelial cell genes (Flt1, Cldn5) despite small cluster size, indicating contamination from other cell types incorrectly grouped within excitatory neurons after initial clustering. In total, we filtered out 637 cells that were identified as either low-quality or contaminated with non-neuronal cell types and recovered 9368 excitatory neurons (see Methods, Figure 2—figure supplement 1A, Supplementary file 1).”